# Variational inference via Wasserstein gradient flows

**Marc Lambert**
DGA, INRIA,
Ecole Normale Supérieure,
PSL Research University
marc.lambert@inria.fr

**Sinho Chewi**
MIT
schewi@mit.edu

**Francis Bach**
INRIA,
Ecole Normale Supérieure,
PSL Research University
francis.bach@inria.fr

**Silvère Bonnabel**
MINES Paris PSL,
Université de la Nouvelle-Calédonie
silvere.bonnabel@minesparis.psl.eu

**Philippe Rigollet**
MIT
rigollet@math.mit.edu

## Abstract

Along with Markov chain Monte Carlo (MCMC) methods, variational inference (VI) has emerged as a central computational approach to large-scale Bayesian inference. Rather than sampling from the true posterior $\pi$, VI aims at producing a simple but effective approximation $\hat{\pi}$ to $\pi$ for which summary statistics are easy to compute. However, unlike the well-studied MCMC methodology, algorithmic guarantees for VI are still relatively less well-understood. In this work, we propose principled methods for VI, in which $\hat{\pi}$ is taken to be a Gaussian or a mixture of Gaussians, which rest upon the theory of gradient flows on the Bures–Wasserstein space of Gaussian measures. Akin to MCMC, it comes with strong theoretical guarantees when $\pi$ is log-concave.

## 1 Introduction

This work brings together three active research areas: variational inference, variational Kalman filtering, and gradient flows on the Wasserstein space.

**Variational inference.** The development of large-scale Bayesian methods has fueled the need for fast and scalable methods to approximate complex distributions. More specifically, Bayesian methodology typically generates a high-dimensional posterior distribution $\pi \propto \exp(-V)$ that is known only up to normalizing constants, making the computation even of simple summary statistics such as the mean and covariance a major computational hurdle. To overcome this limitation, two distinct computational approaches are largely favored. The first approach consists of Markov chain Monte Carlo (MCMC) methods that rely on carefully constructed Markov chains which (approximately) converge to $\pi$. For example, the *Langevin diffusion*

$$\mathrm{d}X_t = -\nabla V(X_t)\,\mathrm{d}t + \sqrt{2}\,\mathrm{d}B_t\,, \tag{1}$$

where $(B_t)_{t \geq 0}$ denotes standard Brownian motion on $\mathbb{R}^d$, admits $\pi$ as a stationary distribution. Crucially, the Langevin diffusion can be discretized and implemented without knowledge of the normalizing constant of $\pi$, leading to practical algorithms for Bayesian inference. Recent theoretical efforts have produced sharp non-asymptotic convergence guarantees for algorithms based on the Langevin diffusion (or variants thereof), with many results known when $\pi$ is strongly log-concave or satisfies isoperimetric assumptions [see, e.g., Durmus et al., 2019, Shen and Lee, 2019, Vempala and Wibisono, 2019, Chen et al., 2020, Dalalyan and Riou-Durand, 2020, Chewi et al., 2021, Lee et al., 2021, Ma et al., 2021, Wu et al., 2022].

36th Conference on Neural Information Processing Systems (NeurIPS 2022).

More recently, Variational Inference (VI) has emerged as a viable alternative to MCMC [Jordan et al., 1999, Wainwright and Jordan, 2008, Blei et al., 2017]. The goal of VI is to approximate the posterior $\pi$ by a more tractable distribution $\hat{\pi} \in \mathcal{P}$ such that

$$\hat{\pi} \in \underset{p \in \mathcal{P}}{\arg\min} \, \mathsf{KL}(p \,\|\, \pi) \, . \tag{2}$$

A common example arises when $\mathcal{P}$ is the class of product distributions, in which case $\hat{\pi}$ is called the *mean-field* approximation of $\mathcal{P}$. Unfortunately, by definition, mean-field approximations fail to capture important correlations present in the posterior $\pi$, and various remedies have been proposed, with varied levels of success. In this paper, we largely focus on obtaining a Gaussian approximation to $\pi$, that is, we take $\mathcal{P}$ to be the class of non-degenerate Gaussian distributions on $\mathbb{R}^d$ [Barber and Bishop, 1997, Seeger, 1999, Honkela and Valpola, 2004, Opper and Archambeau, 2009, Zhang et al., 2018]. The expressive power of the variational model may then be further increased by considering mixture distributions [Lin et al., 2019, Daudel and Douc, 2021, Daudel et al., 2021].

Although the solution $\hat{\pi}$ of (2) is no longer equal to the true posterior, variational inference remains heavily used in practice because the problem (2) can be solved for simple models $\mathcal{P}$ via scalable optimization algorithms. In particular, VI avoids many of the practical hurdles associated with MCMC methods—such as the potentially long "burn-in" period of samplers and the lack of effective stopping criteria for the algorithm—while still producing informative summary statistics. In this regard, we highlight the fact that obtaining an approximation for the covariance matrix of $\pi$ via MCMC methods requires drawing potentially many samples, whereas for many choices of $\mathcal{P}$ (e.g., the Gaussian approximation) the covariance matrix of $\hat{\pi}$ can be directly obtained from the solution to the VI problem (2).

However, in contrast with MCMC methods, to date there have not been many theoretical guarantees for VI, even when $\pi$ is strongly log-concave and $\mathcal{P}$ is taken to be the class of Gaussians $\mathcal{N}(m, \Sigma)$. The problem stems from the fact that the objective in (2) is typically non-convex in the pair $(m, \Sigma)$. Obtaining such guarantees remains a pressing challenge for the field.

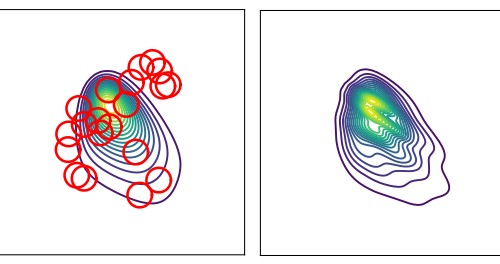

Figure 1: Left: randomly initialized mixture of 20 Gaussians (the initial covariances are depicted as red circles) and contour plot of a logistic target $\pi$. Right: contour lines of a mixture of Gaussians approximation $\hat{\pi}$ obtained from the gradient flow in Section 5.

**Variational Kalman filtering.** There is also considerable interest in extending ideas behind variational inference to dynamical settings of Bayesian inference. Consider a general framework where $(\pi_t)_t$ represents the marginal laws of a stochastic process indexed by time $t$, which can be discrete or continuous. The goal is to recursively build a Gaussian approximation to $(\pi_t)_t$.

As a concrete example, suppose that $(\pi_t)_{t \geq 0}$ denotes the marginal law of the solution to the Langevin diffusion (1). In the context of Bayesian optimal filtering and smoothing, Särkkä [2007] proposed the following heuristic. Let $(m_t, \Sigma_t)$ denote the mean and covariance matrix of $\pi_t$. Then, it can be checked (see Section B.4) that

$$\dot{m}_t = -\, \mathbb{E}\, \nabla V(X_t)$$
$$\dot{\Sigma}_t = 2I - \mathbb{E}[\nabla V(X_t) \otimes (X_t - m_t) + (X_t - m_t) \otimes \nabla V(X_t)] \tag{3}$$

where $X_t \sim \pi_t$. These ordinary differential equations (ODEs) are intractable because they involve expectations under the law of $X_t \sim \pi_t$, which is not available to the practitioner. However, if we replace $X_t \sim \pi_t$ with a Gaussian $Y_t \sim p_t = \mathcal{N}(m_t, \Sigma_t)$ with the same mean and covariance as $X_t$, then the system of ODEs

$$\boxed{\begin{aligned} \dot{m}_t &= -\, \mathbb{E}\, \nabla V(Y_t) \\ \dot{\Sigma}_t &= 2I - \mathbb{E}[\nabla V(Y_t) \otimes (Y_t - m_t) + (Y_t - m_t) \otimes \nabla V(Y_t)] \end{aligned}} \tag{4}$$

yields a well-defined evolution of Gaussian distributions $(p_t)_{t \geq 0}$, which we may optimistically believe to be a good approximation of $(\pi_t)_{t \geq 0}$. Moreover, the system of ODEs can be numerically approximated efficiently in practice using Gaussian quadrature rules to compute the above expectations. This is the principle behind the unscented Kalman filter [Julier et al., 2000].

In the context of the Langevin diffusion, Särkkä's heuristic (4) provides a promising avenue towards computational VI. Indeed, since $\pi \propto \exp(-V)$ is the unique stationary distribution of the Langevin diffusion (1), an algorithm to approximate $(\pi_t)_{t \geq 0}$ is expected to furnish an algorithm to solve the VI problem (2). However, at present there is little theoretical understanding of how the system (4) approximates (3); moreover, Särkkä's heuristic only provides Gaussian approximations, and it is unclear how to extend the system (4) to more complex models (e.g., mixtures of Gaussians).

**Our contributions: bridging the gap via Wasserstein gradient flows.** We show that the approximation $(p_t)_{t \geq 0}$ in Särkkä's heuristic (4) arises precisely as the gradient flow of the Kullback–Leibler (KL) divergence $\mathsf{KL}(\cdot \,\|\, \pi)$ on the Bures–Wasserstein space of Gaussian distributions on $\mathbb{R}^d$ endowed with the 2-Wasserstein distance from optimal transport [Villani, 2003]. This perspective allows us to not only understand its convergence but also to extend it to the richer space of mixtures of Gaussian distributions, and propose an implementation as a novel system of interacting "Gaussian particles". Below, we proceed to describe our contributions in greater detail.

Our framework builds upon the seminal work of Jordan et al. [1998], which introduced the celebrated *JKO scheme* in order to give meaning to the idea that the evolving marginal law of the Langevin diffusion (1) is a gradient flow of $\mathsf{KL}(\cdot \,\|\, \pi)$ on the Wasserstein space $\mathcal{P}_2(\mathbb{R}^d)$ of probability measures with finite second moments. Subsequently, in order to emphasize the Riemannian geometry underlying this result, Otto [2001] developed his eponymous calculus on $\mathcal{P}_2(\mathbb{R}^d)$, a framework which has had tremendous impact in analysis, geometry, PDE, probability, and statistics.

Inspired by this perspective, we show in Theorem 1 that Särkkä's approximation $(p_t)_{t \geq 0}$ is also a gradient flow of $\mathsf{KL}(\cdot \,\|\, \pi)$, with the main difference being that it is *constrained* to lie on the submanifold $\mathsf{BW}(\mathbb{R}^d)$ of $\mathcal{P}_2(\mathbb{R}^d)$ consisting of Gaussian distributions, known as the Bures–Wasserstein manifold. In turn, our result paves the way for new theoretical understanding via the powerful theory of gradient flows. As a first step, using well-known results about convex functionals on the Wasserstein space, we show in Corollary 1 that $(p_t)_{t \geq 0}$ converges rapidly to the solution of the VI problem (2) with $\mathcal{P} = \mathsf{BW}(\mathbb{R}^d)$ as soon as $V$ is convex. Moreover, in Section 4.1, we apply numerical integration based on cubature rules for Gaussian integrals to the system of ODEs (4), thus arriving at a fast method with robust empirical performance (details in Sections I and J).

This combination of results brings VI closer to Langevin-based MCMC both on the practical and theoretical fronts, but still falls short of achieving non-asymptotic discretization guarantees as pioneered by Dalalyan [2017] for MCMC. To further close the theoretical gap between VI and the state of the art for MCMC, we propose in Section 4.2 a stochastic gradient descent (SGD) algorithm as a time discretization of the Bures–Wasserstein gradient flow. This algorithm comes with convergence guarantees that establish VI as a solid competitor to MCMC not only from a practical standpoint but also from a theoretical one. Both have their relative merits; whereas MCMC targets the true posterior, VI leads to fast computation of summary statistics of the approximation $\hat{\pi}$ to $\pi$.

In Section 5, we consider an extension of these ideas to the substantially more flexible class of mixtures of Gaussians. Namely, the space of mixtures of Gaussians can be identified as a Wasserstein space over $\mathsf{BW}(\mathbb{R}^d)$ and hence inherits Otto's differential calculus. Leveraging this viewpoint, in Theorem 3 we derive the gradient flow of $\mathsf{KL}(\cdot \,\|\, \pi)$ over the space of mixtures of Gaussians and propose to implement it via a system of interacting particles. Unlike typical particle-based algorithms, here our particles correspond to Gaussian distributions, and the collection thereof to a Gaussian mixture which is better equipped to approximate a continuous measure. We validate the empirical performance of our method with promising experimental results (see Section J). Although we focus on the VI problem in this work, we anticipate that our notion of "Gaussian particles" may be a broadly useful extension of classical particle methods for PDEs.

**Related work.** Classical VI methods define a parametric family $\mathcal{P} = \{p_\theta : \theta \in \Theta\}$ and minimize $\theta \mapsto \mathsf{KL}(p_\theta \,\|\, \pi)$ over $\theta \in \Theta$ using off-the-shelf optimization algorithms [Paisley et al., 2012, Ranganath et al., 2014]. Since (2) is an optimization problem over the space of probability distributions, we argue for methods that respect a natural geometric structure on this space. In this regard, previous approaches to VI using natural gradients implicitly employ a different geometry [Wu et al., 2019, Huang et al., 2022, Khan and Håvard, 2022], namely the reparameterization-invariant Fisher–Rao geometry [Amari and Nagaoka, 2000]. The application of Wasserstein gradient flows to VI was

introduced earlier in work on normalizing flows and Stein Variational Gradient Descent (SVGD) [Liu and Wang, 2016, Liu, 2017].

Our work falls in line with a number of recent papers aiming to place VI on a solid theoretical footing [Alquier et al., 2016, Wang and Blei, 2019, Domke, 2020, Knoblauch et al., 2022]. Some of these works in particular have obtained non-asymptotic algorithmic guarantees for specific examples, see, e.g., Challis and Barber [2013], Alquier and Ridgway [2020].

The connection between VI and Kalman filtering was studied in the static case by Lambert et al. [2021, 2022a], and extended to the dynamical case by Lambert et al. [2022b], providing a first justification of Särkkä's heuristic in terms of local variational Gaussian approximation. In particular, the closest linear process to the Langevin diffusion (1) is a Gaussian process governed by a McKean–Vlasov equation whose Gaussian marginals have parameters evolving according to Särkkä's ODEs.

Constrained gradient flows on the Wasserstein space have also been extensively studied [Carlen and Gangbo, 2003, Caglioti et al., 2009, Tudorascu and Wunsch, 2011, Eberle et al., 2017], although our interpretation of Särkkä's heuristic is, to the best of our knowledge, new.

## 2 Background

In order to define gradient flows on the space of probability measures, we must first endow this space with a geometry; see Appendix B for more details. Given probability measures $\mu$ and $\nu$ on $\mathbb{R}^d$, define the 2-*Wasserstein distance*

$$W_2(\mu, \nu) = \left[ \inf_{\gamma \in \mathcal{C}(\mu, \nu)} \int \|x - y\|^2 \, \mathrm{d}\gamma(x, y) \right]^{1/2},$$

where $\mathcal{C}(\mu, \nu)$ is the set of *couplings* of $\mu$ and $\nu$, that is, joint distributions on $\mathbb{R}^d \times \mathbb{R}^d$ whose marginals are $\mu$ and $\nu$ respectively. This quantity is finite as long as $\mu$ and $\nu$ belong to the space $\mathcal{P}_2(\mathbb{R}^d)$ of probability measures over $\mathbb{R}^d$ with finite second moments. The 2-Wasserstein distance has the interpretation of measuring the smallest possible mean squared displacement of mass required to *transport* $\mu$ to $\nu$; we refer to Villani [2003, 2009], Santambrogio [2015] for textbook treatments on optimal transport. Unlike other notions of distance between probability measures, such as the total variation distance, the 2-Wasserstein distance respects the geometry of the underlying space $\mathbb{R}^d$, leading to numerous applications in modern data science [see, e.g., Peyré and Cuturi, 2019].

The space $(\mathcal{P}_2(\mathbb{R}^d), W_2)$ is a metric space [Villani, 2003, Theorem 7.3], and we refer to it as the *Wasserstein space*. However, as shown by Otto [Otto, 2001], it has a far richer geometric structure: formally, $(\mathcal{P}_2(\mathbb{R}^d), W_2)$ can be viewed as a Riemannian manifold, a fact which allows for considering gradient flows of functionals on $\mathcal{P}_2(\mathbb{R}^d)$. A fundamental example of such a functional is the KL divergence $\mathsf{KL}(\cdot \parallel \pi)$ to a target density $\pi \propto \exp(-V)$ on $\mathbb{R}^d$, for which Jordan et al. [1998] showed that the Wasserstein gradient flow is the same as the evolution of the marginal law of the Langevin diffusion (1). This optimization perspective has had tremendous impact on our understanding and development of MCMC algorithms [Wibisono, 2018].

## 3 Variational inference with Gaussians

In this section we describe our problem using two equivalent approaches: a variational approach based on a modified version of the JKO scheme of Jordan et al. [1998] (Section 3.1), and a Wasserstein gradient flow approach based on Otto calculus (Section 3.2). Both lead to the same result (Section 3.3). While the former is more accessible to readers who are unfamiliar with gradient flows on the Wasserstein space, the latter leads to strong convergence guarantees (Section 3.4).

### 3.1 Variational approach: the Bures–JKO scheme

The space of non-degenerate Gaussian distributions on $\mathbb{R}^d$ equipped with the $W_2$ distance forms the *Bures–Wasserstein space* $\mathsf{BW}(\mathbb{R}^d) \subseteq \mathcal{P}_2(\mathbb{R}^d)$. On $\mathsf{BW}(\mathbb{R}^d)$, the Wasserstein distance $W_2^2(p_0, p_1)$ between two Gaussians $p_0 = \mathcal{N}(m_0, \Sigma_0)$ and $p_1 = \mathcal{N}(m_1, \Sigma_1)$ admits the following closed form:

$$W_2^2(p_0, p_1) = \|m_0 - m_1\|^2 + \mathcal{B}^2(\Sigma_0, \Sigma_1), \tag{5}$$

where $\mathcal{B}^2(\Sigma_0, \Sigma_1) = \mathrm{tr}(\Sigma_0 + \Sigma_1 - 2\,(\Sigma_0^{\frac{1}{2}} \Sigma_1 \Sigma_0^{\frac{1}{2}})^{\frac{1}{2}})$ is the squared Bures metric [Bures, 1969].

Given a target density $\pi \propto \exp(-V)$ on $\mathbb{R}^d$, and with a step size $h > 0$, we may define the iterates of the proximal point algorithm

$$p_{k+1,h} := \underset{p \in \mathsf{BW}(\mathbb{R}^d)}{\arg\min} \left\{ \mathsf{KL}(p \parallel \pi) + \frac{1}{2h} W_2^2(p, p_{k,h}) \right\}. \tag{6}$$

Using (5), this is an explicit optimization problem involving the mean and covariance matrix of $p$. Although (6) is not solvable in closed form, by letting $h \searrow 0$ we obtain a limiting curve $(p_t)_{t \geq 0}$ via $p_t = \lim_{h \searrow 0} p_{\lfloor t/h \rfloor, h}$, which can be interpreted as the Bures–Wasserstein gradient flow of the KL divergence $\mathsf{KL}(\cdot \parallel \pi)$. This procedure mimics the JKO scheme [Jordan et al., 1998] with the additional constraint that the iterates lie in $\mathsf{BW}(\mathbb{R}^d)$, and we therefore call it the Bures–JKO scheme.

## 3.2 Geometric approach: the Bures–Wasserstein gradient flow of the KL divergence

In the formal sense of Otto described above, $\mathsf{BW}(\mathbb{R}^d)$ is a submanifold of $\mathcal{P}_2(\mathbb{R}^d)$. Moreover, since Gaussians can be parameterized by their mean and covariance, $\mathsf{BW}(\mathbb{R}^d)$ can be identified with the manifold $\mathbb{R}^d \times \mathbf{S}_{++}^d$, where $\mathbf{S}_{++}^d$ is the cone of symmetric positive definite $d \times d$ matrices. Hence, $\mathsf{BW}(\mathbb{R}^d)$ is a genuine Riemannian manifold in its own right [see Modin, 2017, Malagò et al., 2018, Bhatia et al., 2019], and gradient flows can be defined using Riemannian geometry [do Carmo, 1992]. See Section B.3 for more details. Since the functional $\mu \mapsto \mathcal{F}(\mu) = \mathsf{KL}(\mu \parallel \pi)$ defined over $\mathcal{P}_2(\mathbb{R}^d)$ restricts to a functional over $\mathsf{BW}(\mathbb{R}^d)$, we can also consider the gradient flow of $\mathcal{F}$ over the Bures–Wasserstein space; note that this latter gradient flow is necessarily a curve $(p_t)_{t \geq 0}$ such that each $p_t$ is a Gaussian measure.

## 3.3 Variational inference via the Bures–Wasserstein gradient flow

Using either approach, we can prove the following theorem.

**Theorem 1.** *Let $\pi \propto \exp(-V)$ be the target density on $\mathbb{R}^d$. Then, the limiting curve $(p_t)_{t \geq 0}$ where $p_t = \mathcal{N}(m_t, \Sigma_t)$ is obtained via the Bures–JKO scheme (6), or equivalently, the Bures–Wasserstein gradient flow $(p_t)_{t \geq 0}$ of the KL divergence $\mathsf{KL}(\cdot \parallel \pi)$, satisfies Särkkä's system of ODEs (4).*

*Proof.* The proof using the Bures–JKO scheme is given in Section A.1 and the proof using Otto calculus is presented in Section C. $\qquad \square$

This theorem shows that Särkkä's heuristic (4) precisely yields the Wasserstein gradient flow of the KL divergence over the submanifold $\mathsf{BW}(\mathbb{R}^d)$. Equipped with this interpretation, we are now able to obtain information about the asymptotic behavior of the approximation $(p_t)_{t \geq 0}$. Namely, we can hope that it converges to constrained minimizer $\hat{\pi} = \arg\min_{p \in \mathsf{BW}(\mathbb{R}^d)} \mathsf{KL}(p \parallel \pi)$, i.e., precisely the solution to the VI problem (2). In the next section, we show that this convergence in fact holds as soon as $V$ is convex, and moreover with quantitative rates.

The solution $\hat{\pi}$ to (2), and consequently the limit point of Särkkä's approximation, is well-studied in the variational inference literature [see, e.g., Opper and Archambeau, 2009], and we recall standard facts about $\hat{\pi}$ here for completeness. It is known that $\hat{\pi}$ satisfies the equations

$$\mathbb{E}_{\hat{\pi}} \nabla V = 0 \qquad \text{and} \qquad \mathbb{E}_{\hat{\pi}} \nabla^2 V = \hat{\Sigma}^{-1}, \tag{7}$$

where $\hat{\Sigma}$ is the covariance matrix of $\hat{\pi}$ (these equations can also be derived as first-order necessary conditions by setting the Bures–Wasserstein gradient derived in Section C to zero). In particular, it follows from (7) that if $\nabla^2 V$ enjoys the bounds $\alpha I \preceq \nabla^2 V \preceq \beta I$ for some $-\infty \leq \alpha \leq \beta \leq \infty$, then any solution $\hat{\pi}$ to the constrained problem also satisfies $\beta^{-1} I \preceq \hat{\Sigma} \preceq (\alpha \vee 0)^{-1} I$.

## 3.4 Continuous-time convergence

Besides providing an intuitive interpretation of Särkkä's heuristic, Theorem 1 readily yields convergence criteria for the system (4) which rest upon general principles for gradient flows. We begin with

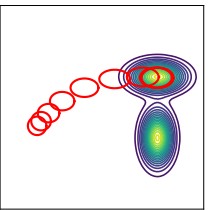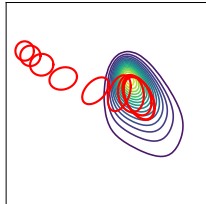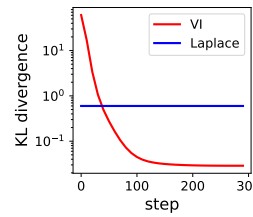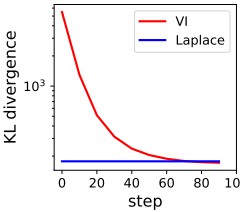

Figure 2: Two left plots: approximation of a bimodal target and a logistic target. Two right plots: convergence of the KL in dimension 2 and 100 for the logistic target. Our algorithm yields better approximation in KL than the Laplace approximation (see Appendix I.4 for details).

a key observation. For a functional $\mathcal{F} : \mathsf{BW}(\mathbb{R}^d) \to \mathbb{R} \cup \{\infty\}$ and $\alpha \in \mathbb{R}$, we say that $\mathcal{F}$ is *α-convex* if for all constant-speed geodesics $(p_t)_{t \in [0,1]}$ in $\mathsf{BW}(\mathbb{R}^d)$,

$$\mathcal{F}(p_t) \le (1-t)\,\mathcal{F}(p_0) + t\,\mathcal{F}(p_1) - \frac{\alpha\,t\,(1-t)}{2}\,W_2^2(p_0, p_1), \qquad t \in [0,1].$$

**Lemma 1.** *For any $\alpha \in \mathbb{R}$, if $\nabla^2 V \succeq \alpha I$, then $\mathsf{KL}(\cdot \,\|\, \pi)$ is α-convex on $\mathsf{BW}(\mathbb{R}^d)$.*

*Proof.* The assumption that $\nabla^2 V \succeq \alpha I$ entails that the functional $\mathsf{KL}(\cdot \,\|\, \pi)$ is α-convex on the entire Wasserstein space $(\mathcal{P}_2(\mathbb{R}^d), W_2)$ [see, e.g., Villani, 2009, Theorem 17.15]. Since $\mathsf{BW}(\mathbb{R}^d)$ is a geodesically convex subset of $\mathcal{P}_2(\mathbb{R}^d)$ (see Section B.3), then the geodesics in $\mathsf{BW}(\mathbb{R}^d)$ agree with the geodesics in $\mathcal{P}_2(\mathbb{R}^d)$, from which it follows that $\mathsf{KL}(\cdot \,\|\, \pi)$ is α-convex on $\mathsf{BW}(\mathbb{R}^d)$. □

Consequently, we obtain the following corollary. Its proof is postponed to Section D.

**Corollary 1.** *Suppose that $\nabla^2 V \succeq \alpha I$ for some $\alpha \in \mathbb{R}$. Then, for any $p_0 \in \mathsf{BW}(\mathbb{R}^d)$, there is a unique solution to the $\mathsf{BW}(\mathbb{R}^d)$ gradient flow of $\mathsf{KL}(\cdot \,\|\, \pi)$ started at $p_0$. Moreover:*

*1. If $\alpha > 0$, then for all $t \ge 0$, $W_2^2(p_t, \hat{\pi}) \le \exp(-2\alpha t)\,W_2^2(p_0, \hat{\pi})$.*

*2. If $\alpha > 0$, then for all $t \ge 0$, $\mathsf{KL}(p_t \,\|\, \pi) - \mathsf{KL}(\hat{\pi} \,\|\, \pi) \le \exp(-2\alpha t)\,\{\mathsf{KL}(p_0 \,\|\, \pi) - \mathsf{KL}(\hat{\pi} \,\|\, \pi)\}.$*

*3. If $\alpha = 0$, then for all $t > 0$, $\mathsf{KL}(p_t \,\|\, \pi) - \mathsf{KL}(\hat{\pi} \,\|\, \pi) \le \frac{1}{2t}\,W_2^2(p_0, \hat{\pi})$.*

The assumption that $\nabla^2 V \succeq \alpha I$ for some $\alpha > 0$, i.e., that $\pi$ is *strongly log-concave*, is a standard assumption in the MCMC literature. Under this same assumption, Corollary 1 yields convergence for the Bures–Wasserstein gradient flow of $\mathsf{KL}(\cdot \,\|\, \pi)$; however, the flow must first be discretized in time for implementation. If we assume additionally that the smoothness condition $\nabla^2 V \preceq \beta I$ holds, then a surge of recent research has succeeded in obtaining precise non-asymptotic guarantees for discretized MCMC algorithms. In Section 4.2 below, we will show how to do the same for VI.

## 4 Time discretization of the Bures–Wasserstein gradient flow

We are now equipped with dual perspectives on a dynamical solution to Gaussian VI: ODE and gradient flow. Each perspective leads to a different implementation. On the one hand, we discretize the system of ODEs defined in (4) using numerical integration. On the other, we discretize the gradient flow using stochastic gradient descent in the Bures–Wasserstein space.

### 4.1 Numerical integration of the ODEs

The system of ODEs (4) can be integrated in time using a classical Runge–Kutta scheme. The expectations under a Gaussian support are approximated by cubature rules used in Kalman filtering [Arasaratnam and Haykin, 2009].

Moreover, a square root version of the ODE is also considered to ensure that covariance matrices remain symmetric and positive. See Appendix I.2 for more details. We have tested our method on a bimodal distribution and on a posterior distribution arising from a logistic regression problem. We observe fast convergence as shown in Figure 2.

## 4.2 Bures–Wasserstein SGD and theoretical guarantees for VI

Although the ODE discretization proposed in the preceding section enjoys strong empirical performance, it is unclear how to quantify its impact on the convergence rates established in Corollary 1. Therefore, we now propose a stochastic gradient descent algorithm over the Bures–Wasserstein space, for which useful analysis tools have been developed [Chewi et al., 2020, Altschuler et al., 2022]. This approach bypasses the use of the system of ODEs (4), and instead discretizes the Bures–Wasserstein gradient flow directly. Under the standard assumption of strong log-concavity and log-smoothness, it leads to an algorithm (Algorithm 1) for approximating $\hat{\pi}$ with provable convergence guarantees.

Algorithm 1 maintains a sequence of Gaussian distributions $(p_k)_{k \in \mathbb{N}}$; here $(m_k, \Sigma_k)$ denote the mean vector and covariance matrix at iteration $k$ (see Section E for a derivation of the algorithm as SGD in the Bures–Wasserstein space). The clipping operator $\mathrm{clip}^{\tau}$, which is introduced purely for the purpose of theoretical analysis, simply truncates the eigenvalues from above; see Section E. Our theoretical result for VI is given as the following theorem, whose proof is deferred to Section E.

---
**Algorithm 1** Bures–Wasserstein SGD

**Require:** strong convexity parameter $\alpha > 0$; step size $h > 0$; mean $m_0$ and covariance matrix $\Sigma_0$
    **for** $k = 1, \ldots, N$ **do**
        draw a sample $\hat{X}_k \sim p_k$
        set $m_{k+1} \leftarrow m_k - h \nabla V(\hat{X}_k)$
        set $M_k \leftarrow I - h \left( \nabla^2 V(\hat{X}_k) - \Sigma_k^{-1} \right)$
        set $\Sigma_k^+ \leftarrow M_k \Sigma_k M_k$
        set $\Sigma_{k+1} \leftarrow \mathrm{clip}^{1/\alpha} \Sigma_k^+$
    **end for**
---

**Theorem 2.** *Assume that $0 \prec \alpha I \preceq \nabla^2 V \preceq I$. Also, assume that $h \leq \frac{\alpha^2}{60}$ and that we initialize Algorithm 1 at a matrix satisfying $\frac{\alpha}{9} I \preceq \Sigma_{\mu_0} \preceq \frac{1}{\alpha} I$. Then, for all $k \in \mathbb{N}$,*

$$\mathbb{E} \, W_2^2(p_k, \hat{\pi}) \leq \exp(-\alpha k h) \, W_2^2(p_0, \hat{\pi}) + \frac{36 d h}{\alpha^2} \, .$$

*In particular, we obtain $\mathbb{E} \, W_2^2(p_k, \hat{\pi}) \leq \varepsilon^2$ provided we set $h \asymp \frac{\alpha^2 \varepsilon^2}{d}$ and the number of iterations to be $k \gtrsim \frac{d}{\alpha^3 \varepsilon^2} \log(W_2(p_0, \hat{\pi})/\varepsilon)$.*

The upper bound $\nabla^2 V \preceq I$ is notationally convenient for our proof but not necessary; in any case, any strongly log-concave and log-smooth density $\pi$ can be rescaled so that the assumption holds.

Theorem 2 is similar in flavor to modern results for MCMC, both in terms of the assumptions (Hessian bounds and query access to the derivatives[1] of $V$) and the conclusion (a non-asymptotic polynomial-time algorithmic guarantee). We hope that such an encouraging result for VI will prompt more theoretical studies aimed at closing the gap between the two approaches.

## 5 Variational inference with mixtures of Gaussians

Thus far, we have shown that the tractability of Gaussians can be readily exploited in the context of Bures–Wasserstein gradient flows and translated into useful results for variational inference. Nevertheless, these results are limited by the lack of expressivity of Gaussians, namely their inability to capture complex features such as multimodality and, more generally, heterogeneity. To overcome this limitation, mixtures of Gaussians arise as a natural and powerful alternative; indeed, universal approximation of arbitrary probability measures by mixtures of Gaussians is well-known [see, e.g., Delon and Desolneux, 2020]. As we show next, the space of mixtures of Gaussians can also be equipped with a Wasserstein structure which gives rise to implementable gradient flows.

### 5.1 Geometry of the space of mixtures of Gaussians

We begin with the key observation already made by Chen et al. [2019], that any mixture of Gaussians can be canonically identified with a probability distribution (the mixing distribution) over the parameter space $\Theta = \mathbb{R}^d \times \mathbf{S}_{++}^d$ (the space of means and covariance matrices). Explicitly a probability measure $\mu \in \mathcal{P}(\Theta)$ corresponds to a Gaussian mixture as follows:

$$\mu \qquad \leftrightarrow \qquad \mathsf{p}_\mu := \int p_\theta \, \mathrm{d}\mu(\theta) \, , \tag{8}$$

---
[1] A notable downside of Algorithm 1 is the requirement of a Hessian oracle for $V$, which results in a higher per-iteration cost than typical MCMC samplers.

where $p_\theta$ is the Gaussian distribution with parameters $\theta \in \Theta$. Equivalently, $\mu$ can be thought of as a probability measure over $\mathsf{BW}(\mathbb{R}^d)$, and hence the space of Gaussian mixtures on $\mathbb{R}^d$ can be identified with the Wasserstein space $\mathcal{P}_2(\mathsf{BW}(\mathbb{R}^d))$ over the Bures–Wasserstein space which is endowed with the distance (5) between Gaussian measures. Indeed, the theory of optimal transport can be developed with any Riemannian manifold (rather than $\mathbb{R}^d$) as the base space [Villani, 2009]. As before, the space $\mathcal{P}_2(\mathsf{BW}(\mathbb{R}^d))$ is endowed with a formal Riemannian structure, which respects the geometry of the base space $\mathsf{BW}(\mathbb{R}^d)$, and we can consider Wasserstein gradient flows over $\mathcal{P}_2(\mathsf{BW}(\mathbb{R}^d))$.

Note that this framework encompasses both discrete mixtures of Gaussians (when $\mu$ is a discrete measure) and continuous mixtures of Gaussians. In the case when the mixing distribution $\mu$ is discrete, the geometry of $\mathcal{P}_2(\mathsf{BW}(\mathbb{R}^d))$ was studied by Chen et al. [2019], Delon and Desolneux [2020]. An important insight of our work, however, is that it is fruitful to consider the full space $\mathcal{P}_2(\mathsf{BW}(\mathbb{R}^d))$ for deriving gradient flows, even if we eventually develop algorithms which propagate a finite number of mixture components.

## 5.2 Gradient flow of the KL divergence and particle discretization

We consider the gradient flow of the KL divergence functional

$$\mu \mapsto \mathcal{F}(\mu) := \mathsf{KL}(\mathsf{p}_\mu \,\|\, \pi) \tag{9}$$

over the space $\mathcal{P}_2(\mathsf{BW}(\mathbb{R}^d))$. The proof of the following theorem is given in Section F.

**Theorem 3.** *The gradient flow $(\mu_t)_{t \geq 0}$ of the functional $\mathcal{F}$ defined in (9) over $\mathcal{P}_2(\mathsf{BW}(\mathbb{R}^d))$ can be described as follows. Let $\theta_0 = (m_0, \bar{\Sigma}_0) \sim \mu_0$, and let $\theta_t = (m_t, \Sigma_t)$ evolve according to the ODE*

$$\boxed{\begin{aligned} \dot{m}_t &= -\,\mathbb{E}\,\nabla \ln \frac{\mathsf{p}_{\mu_t}}{\pi}(Y_t) \\ \dot{\Sigma}_t &= -\,\mathbb{E}\,\nabla^2 \ln \frac{\mathsf{p}_{\mu_t}}{\pi}(Y_t)\,\Sigma_t - \Sigma_t\,\mathbb{E}\,\nabla^2 \ln \frac{\mathsf{p}_{\mu_t}}{\pi}(Y_t) \end{aligned}} \tag{10}$$

*where $Y_t \sim \mathcal{N}(m_t, \Sigma_t)$. Then $\theta_t \sim \mu_t$.*

The gradient flow in Theorem 3 describes the evolution of a particle $\theta_t$ which describes the parameters of a Gaussian measure, hence the name *Gaussian particle*. The intuition behind this evolution is as follows. Suppose we draw infinitely many initial particles (each being a Gaussian) from $\mu_0$. By evolving all those particles through (10), which interact with each other via the term $\mathsf{p}_{\mu_t}$, they tend to aggregate in some parts of the space of Gaussian parameters and spread out in others. This distribution of Gaussian particles is precisely the mixing measure $\mu_t$, which, in turn, corresponds to a Gaussian mixture. Since an infinite number of Gaussian particles is impractical, consider initializing this evolution at a finitely supported distribution $\mu_0$, thus corresponding to a more familiar Gaussian mixture model with a finite number of components:

$$\mu_0 = \frac{1}{N}\sum_{i=1}^{N}\delta_{\theta_0^{(i)}} = \frac{1}{N}\sum_{i=1}^{N}\delta_{(m_0^{(i)}, \Sigma_0^{(i)})} \qquad \leftrightarrow \qquad \mathsf{p}_{\mu_0} := \frac{1}{N}\sum_{i=1}^{N}p_{(m_0^{(i)}, \Sigma_0^{(i)})}\,.$$

Interestingly, it can be readily checked that the system of ODEs (10) thus initialized maintains a finite mixture distribution:

$$\mu_t = \frac{1}{N}\sum_{i=1}^{N}\delta_{\theta_t^{(i)}} = \frac{1}{N}\sum_{i=1}^{N}\delta_{(m_t^{(i)}, \Sigma_t^{(i)})}\,,$$

where the parameters $\theta_t^{(i)} = (m_t^{(i)}, \Sigma_t^{(i)})$ evolve according to the following interacting particle system, for $i \in [N]$

$$\dot{m}_t^{(i)} = -\,\mathbb{E}\,\nabla \ln \frac{\mathsf{p}_{\mu_t}}{\pi}(Y_t^{(i)})\,, \tag{11}$$

$$\dot{\Sigma}_t^{(i)} = -\,\mathbb{E}\,\nabla^2 \ln \frac{\mathsf{p}_{\mu_t}}{\pi}(Y_t^{(i)})\,\Sigma_t^{(i)} - \Sigma_t^{(i)}\,\mathbb{E}\,\nabla^2 \ln \frac{\mathsf{p}_{\mu_t}}{\pi}(Y_t^{(i)})\,, \tag{12}$$

where $Y_t^{(i)} \sim p_{\theta_t^{(i)}}$. This finite system of particles can now be implemented using the same numerical tools as for Gaussian VI, see Section J. Note that due to this property of the dynamics, we can hope at

best to converge to the best mixture of $N$ Gaussians approximating $\pi$, but this approximation error is expected to vanish as $N \to \infty$. Also, similarly to (4), it is possible to write down Hessian-free updates using integration by parts, see Appendix A.2.

The above system of particles may also be derived using a proximal point method similar to the Bures–JKO scheme, see Section A.2. Indeed, infinitesimally, it has the variational interpretation

$$(\theta_{t+h}^{(1)}, \ldots, \theta_{t+h}^{(N)}) \approx \operatorname*{arg\,min}_{\theta^{(1)}, \ldots, \theta^{(N)} \in \Theta} \left\{ \mathsf{KL}\Big( \frac{1}{N} \sum_{i=1}^{N} p_{\theta^{(i)}} \,\Big\|\, \pi \Big) + \frac{1}{2Nh} \sum_{i=1}^{N} W_2^2(p_{\theta^{(i)}}, p_{\theta_t^{(i)}}) \right\}.$$

Reassuringly, Equations (11)-(12) reduce to (4) when $\mu_0 = \delta_{(m_0, \Sigma_0)}$ is a point mass, indicating that the theorem provides a natural extension of our previous results. However, although the model (8) is substantially more expressive than the Gaussian VI considered in Section 3, it has the downside that we lose many of the theoretical guarantees. For example, even when $V$ is convex, the objective functional $\mathcal{F}$ considered here need not be convex; see Section G. We nevertheless validate the practical utility of our approach in experiments (see Figure 3 and Section J).

Unlike typical interacting particle systems which arise from discretizations of Wasserstein gradient flows, at each time $t$, the distribution $\mathsf{p}_{\mu_t}$ is continuous. This extension provides considerably more flexibility—from a mixture of point masses to a mixture of Gaussians—compared to interacting particle-based algorithms hitherto considered for either sampling [Liu and Wang, 2016, Liu, 2017, Duncan et al., 2019, Chewi et al., 2020], or solving partial differential equations [Carrillo et al., 2011, 2012, Bonaschi et al., 2015, Craig and Bertozzi, 2016, Carrillo et al., 2019, Craig et al., 2022].

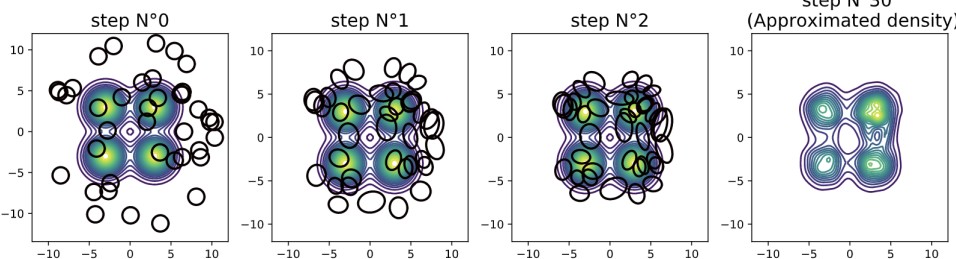

Figure 3: Approximation of a Gaussian mixture target $\pi$ with 40 Gaussian particles. The particles are represented by their covariance ellipsoids shown at Steps 0, 1, and 2. The right figure shows the final step with the approximated density in contour-lines. More figures are available in Appendix J.

## 6   Conclusion

Using the powerful theory of Wasserstein gradient flows, we derived new algorithms for VI using either Gaussians or mixtures of Gaussians as approximating distributions. The consequences are twofold. On the one hand, strong convergence guarantees under classical conditions contribute markedly to closing the theoretical gap between MCMC and Gaussian VI. On the other hand, discretization of the Wasserstein gradient flow for mixtures of Gaussians yields a new *Gaussian particle method* for time discretization which, unlike classical particle methods, maintains a continuous probability distribution at each time.

We conclude by briefly listing some possible directions for future study. For Gaussian variational inference, our theoretical result (Theorem 2) can be strengthened by weakening the assumption that $\pi$ is strongly log-concave, or by developing algorithms which do not require Hessian information for $V$. For mixtures of Gaussians, it is desirable to design a principled algorithm which also allows for the mixture weights to be updated.

Towards the latter question, in Section H we derive the gradient flow of the KL divergence with respect to the Wasserstein–Fisher–Rao geometry [Liero et al., 2016, Chizat et al., 2018, Liero et al., 2018], which yields an interacting system of Gaussian particles with changing weights. The equations

are given as follows: at each time $t$, the mixing measure is the discrete measure

$$\mu_t = \sum_{i=1}^{N} w_t^{(i)} \delta_{(m_t^{(i)}, \Sigma_t^{(i)})}.$$

Let $Y_t^{(i)} \sim \mathcal{N}(m_t^{(i)}, \Sigma_t^{(i)})$, and let $r_t^{(i)} = \sqrt{w_t^{(i)}}$. Then, the system of ODEs is given by

$$\dot{m}_t^{(i)} = -\mathbb{E}\, \nabla \ln \frac{\mathsf{p}_{\mu_t}}{\pi}(Y_t^{(i)}),$$

$$\dot{\Sigma}_t^{(i)} = -\mathbb{E}\, \nabla^2 \ln \frac{\mathsf{p}_{\mu_t}}{\pi}(Y_t^{(i)})\, \Sigma_t^{(i)} - \Sigma_t^{(i)}\, \mathbb{E}\, \nabla^2 \ln \frac{\mathsf{p}_{\mu_t}}{\pi}(Y_t^{(i)}),$$

$$\dot{r}_t^{(i)} = -\left( \mathbb{E} \ln \frac{\mathsf{p}_{\mu_t}}{\pi}(Y_t^{(i)}) - \frac{1}{N} \sum_{j=1}^{N} \mathbb{E} \ln \frac{\mathsf{p}_{\mu_t}}{\pi}(Y_t^{(j)}) \right) r_t^{(i)}.$$

We have implemented these equations and their empirical performance is encouraging. However, a fuller investigation of algorithms for VI with changing weights is beyond the scope of this work and we leave it for future research.

Code for the experiments is available at https://github.com/marc-h-lambert/W-VI.

## Acknowledgments and Disclosure of Funding

We thank Yian Ma for helpful discussions, as well as anonymous reviewers for useful references and suggestions. ML acknowledges support from the French Defence procurement agency (DGA). SC is supported by the Department of Defense (DoD) through the National Defense Science & Engineering Graduate Fellowship (NDSEG) Program. FB and ML acknowledge support from the French government under the management of the Agence Nationale de la Recherche as part of the "Investissements d'avenir" program, reference ANR-19-P3IA-0001 (PRAIRIE 3IA Institute), as well as from the European Research Council (grant SEQUOIA 724063). PR is supported by NSF grants IIS-1838071, DMS-2022448, and CCF-2106377.

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
