# A   Proofs via the Bures–JKO scheme

## A.1   Proof of Theorem 1

Given a Gaussian distribution at time $t$ written $p_t = \mathcal{N}(m_t, \Sigma_t)$ and a target distribution $\pi$, we seek the solution $p$ at time $t + h$ of the following JKO scheme, where $p = \mathcal{N}(m, \Sigma)$ is constrained to lie on the space $\mathsf{BW}(\mathbb{R}^d)$ of Gaussians equipped with the Wasserstein distance:

$$\min_{p \in \mathsf{BW}(\mathbb{R}^d)} L(p) = \mathsf{KL}(p \parallel \pi) + \frac{1}{2h} W_2^2(p, p_t). \tag{13}$$

Using the expression for the Wasserstein distance $W_2^2(p_t, p)$ given in (5) it is equivalent to finding the Gaussian parameters which solve:

$$\min_{m, \Sigma} L(m, \Sigma) = \mathsf{KL}(\mathcal{N}(m, \Sigma) \parallel \pi) + \frac{1}{2h} \|m_t - m\|^2 + \frac{1}{2h} \mathcal{B}^2(\Sigma_t, \Sigma). \tag{14}$$

We first compute the critical points of $L$ and then take the limit as $h \searrow 0$ to get the desired differential equations (ODEs) for the parameters. This boils down to computing the Wasserstein gradient flow of $\mathsf{KL}(\cdot \parallel \pi)$ over the Bures–Wasserstein manifold $\mathsf{BW}(\mathbb{R}^d)$.

The left KL divergence is a sum of two terms $\mathsf{KL}(p\|\pi) = H(p) - \mathbb{E}_p[\ln \pi]$, where $H(p)$ is the negative entropy of a Gaussian. It satisfies $\nabla_m H(p) = 0$ and $\nabla_\Sigma H(p) = \nabla_\Sigma(-\frac{1}{2} \ln \det \Sigma) = -\frac{1}{2} \Sigma^{-1}$.

To alleviate notation, for any function $f$ we both let $x$ denote its argument and $\mathbb{E}_p[f(x)]$ denote expectation over $x \sim p = \mathcal{N}(m, \Sigma)$ throughout the present proof, depending on the context.

The gradient of the left KL divergence with respect to $m$ is given by:

$$\nabla_m \mathsf{KL}(p \parallel \pi) = -\nabla_m \mathbb{E}_p[\ln \pi(x)] = -\mathbb{E}_p[\nabla_x \ln \pi(x)],$$

where we have used integration by parts (assuming $\pi$ is continuously differentiable) and the property of Gaussian densities $\nabla_m \mathcal{N}(x \mid m, \Sigma) = -\nabla_x \mathcal{N}(x \mid m, \Sigma)$ to get a derivative with respect to $x$. The critical point of $L$ given by (14) w.r.t. the mean parameter $m$ thus writes:

$$\nabla_m L(m, \Sigma) = \frac{1}{h}(m - m_t) - \mathbb{E}_p[\nabla_x \ln \pi(x)] = 0. \tag{15}$$

Taking the limit as $h \searrow 0$, we find that $m_t$ must satisfy the following ODE:

$$\dot{m}_t = \mathbb{E}_{p_t}[\nabla_x \ln \pi(x)] = -\mathbb{E}_{p_t}[\nabla_x V(x)],$$

where we recall that $\pi \propto \exp(-V)$. This recovers the first line of (4).

The gradient of the left KL divergence with respect to $\Sigma$ is given by:

$$\nabla_\Sigma \mathsf{KL}(p \parallel \pi) = -\frac{1}{2} \Sigma^{-1} - \nabla_\Sigma \mathbb{E}_p[\ln \pi(x)] = -\frac{1}{2} \Sigma^{-1} - \frac{1}{2} \mathbb{E}_p[\nabla_x^2 \ln \pi(x)],$$

where we have used two integrations by parts (supposing $\pi$ is twice continuously differentiable) and the property of Gaussian densities $\nabla_\Sigma \mathcal{N}(x \mid m, \Sigma) = \frac{1}{2} \nabla_x^2 \mathcal{N}(x \mid m, \Sigma)$ to let a Hessian w.r.t. $x$ appear. The Bures derivative is given by [see Bhatia et al., 2019]:

$$\nabla_\Sigma \mathcal{B}^2(\Sigma_t, \Sigma) = I - T^{\Sigma, \Sigma_t},$$

where $T^{A,B}$ is the optimal transport map from $\mathcal{N}(0, A)$ to $\mathcal{N}(0, B)$, with the explicit expression $T^{A,B} = A^{-\frac{1}{2}}(A^{\frac{1}{2}} B A^{\frac{1}{2}})^{\frac{1}{2}} A^{-\frac{1}{2}} = (T^{B,A})^{-1}$. The gradient of the variational loss $L$ in (14) is thus:

$$\nabla_\Sigma L(m, \Sigma) = \frac{1}{2h} I - \frac{1}{2h} T^{\Sigma, \Sigma_t} - \frac{1}{2} \Sigma^{-1} - \frac{1}{2} \mathbb{E}_p[\nabla_x^2 \ln \pi(x)].$$

Zeroing this equation gives:

$$I = T^{\Sigma, \Sigma_t} + h \Sigma^{-1} + h \mathbb{E}_p[\nabla_x^2 \ln \pi(x)]. \tag{16}$$

Multiplying by $\Sigma$ on the left, as well as on the right, yields the two following equations:

$$\Sigma = \Sigma T^{\Sigma, \Sigma_t} + hI + h \Sigma \mathbb{E}_p[\nabla_x^2 \ln \pi(x)], \tag{17}$$

$$\Sigma = T^{\Sigma, \Sigma_t} \Sigma + hI + h \mathbb{E}_p[\nabla_x^2 \ln \pi(x)] \Sigma. \tag{18}$$

Averaging the equations we obtain the symmetrized form:

$$\Sigma = \frac{1}{2}\, T^{\Sigma,\Sigma_t}\Sigma + \frac{1}{2}\,\Sigma T^{\Sigma,\Sigma_t} + hI + \frac{1}{2}\, h\,\Sigma\, \mathbb{E}_p[\nabla_x^2 \ln \pi(x)] + \frac{1}{2}\, h\, \mathbb{E}_p[\nabla_x^2 \ln \pi(x)]\,\Sigma\,. \qquad (19)$$

Let us denote $T^{\Sigma,\Sigma_t} = T(\Sigma)$. Since $T(\Sigma)$ pushes forward $\Sigma$ to $\Sigma_t$, it follows that $T(\Sigma)\,\Sigma\, T(\Sigma) = \Sigma_t$ (which can be checked directly from the expression for $T^{\Sigma,\Sigma_t}$). The first variation of this equality w.r.t. $\Sigma$ at $I$ gives

$$dT\,\Sigma_t + d\Sigma + \Sigma_t\, dT = 0\,. \qquad (20)$$

Let us now term $\Sigma = \Sigma_{t+h}$ the solution to (19). Up to the first order in $h$ we have $\Sigma_{t+h} = \Sigma_t + d\Sigma = \Sigma_t + h\dot{\Sigma}_t$. Let $dT$ denote the corresponding first variation of $T$, that is, $T(\Sigma_{t+h}) = T(\Sigma_t) + dT = I + dT$ up to the first order in $h$. Substituting into (19), using the previously found relation (20), dividing by $h$ and letting $h \searrow 0$, we finally obtain the desired ODE:

$$\dot{\Sigma}_t = 2I + \Sigma_t\, \mathbb{E}_{p_t}[\nabla_x^2 \ln \pi(x)] + \mathbb{E}_{p_t}[\nabla_x^2 \ln \pi(x)]\,\Sigma_t \qquad (21)$$

$$= 2I + \mathbb{E}_{p_t}[\nabla_x \ln \pi(x) \otimes (x - m_t)] + \mathbb{E}_{p_t}[(x - m_t) \otimes \nabla_x \ln \pi(x)]\,, \qquad (22)$$

where the relation $\mathbb{E}_p[\nabla_x^2 \ln \pi(x)]\,\Sigma = \mathbb{E}_p[\nabla_x \ln \pi(x) \otimes (x - m)]$ comes from Gaussian integration by parts and yields a Hessian-free form. Letting $\pi \propto \exp(-V)$ yields the second line of (4).

**Interpretation in terms of Wasserstein gradient flows.** Let $T_{t+h \to t}$ denote the optimal transport map from $p_{t+h}$ to $p_t$, so that $T_{t+h \to t} = m_t + T^{\Sigma_{t+h},\Sigma_t}(x - m_{t+h})$. Combining the equations (15) and (16), it reads

$$\frac{T_{t+h \to t}(x) - x}{h} = \frac{1}{h}\left\{m_t - m_{t+h} + (T^{\Sigma_{t+h},\Sigma_t} - I)(x - m_{t+h})\right\}$$

$$= \mathbb{E}_{p_{t+h}}\nabla V + (\mathbb{E}_{p_{t+h}}\nabla^2 V - \Sigma_{t+h}^{-1})(x - m_{t+h})\,.$$

In Section C, this equality will be written

$$\frac{T_{t+h \to t} - \mathrm{id}}{h} = [\nabla_{\mathsf{BW}}\,\mathsf{KL}(\cdot \,\|\, \pi)](p_{t+h})\,, \qquad (23)$$

where $\frac{1}{h}(T_{t+h \to t} - \mathrm{id})$ and $[\nabla_{\mathsf{BW}}\,\mathsf{KL}(\cdot \,\|\, \pi)](p_{t+h})$ are the Bures–Wasserstein gradients of the functionals $-\frac{1}{2h} W_2^2(\cdot, p_t)$ and $\mathsf{KL}(\cdot \,\|\, \pi)$ at $p_{t+h}$ respectively. The equation (23) is a first-order optimality condition for the Bures–JKO scheme (6) and mimics the known optimality condition for the original JKO scheme, see [Santambrogio, 2015, equation (8.4)].

The quantity $\frac{1}{h}(T_{t+h \to t} - \mathrm{id})$ is a difference quotient which measures the infinitesimal displacement of a particle traveling along the gradient flow. As $h \searrow 0$, we will interpret this quantity as $-v_t$, the negative of the *tangent vector* to the curve at time $t$ (the negative sign appears because $T_{t+h \to t}$ is the transport map *backwards* in time). Hence, the equation (23) states that as $h \searrow 0$, the tangent vector to the curve $(p_t)_{t \geq 0}$ is the negative Bures–Wasserstein gradient of the KL divergence, which is the definition of a gradient flow.

From this perspective, the computation of the linearization in (20) is equivalent to computing the tangent vector to the Wasserstein geodesic, which is given in (32).

### A.2 Extension to mixtures of Gaussians

We now consider a finite Gaussian mixture model $p = \frac{1}{N}\sum_{i=1}^{N} p_{\theta^{(i)}}$ where $\theta^{(i)} = (m^{(i)}, \Sigma^{(i)})$. We consider the following variational problem:

$$\min_{\theta^{(1)},\ldots,\theta^{(N)} \in \Theta} \frac{1}{2Nh}\sum_{i=1}^{N} W_2^2(p_{\theta^{(i)}}, p_{\theta_t^{(i)}}) + \mathsf{KL}\left(\frac{1}{N}\sum_{i=1}^{N} p_{\theta^{(i)}} \,\Big\|\, \pi\right),$$

where, as before, $W_2$ is the Wasserstein distance between two Gaussians distribution:

$$W_2^2(p_{\theta^{(i)}}, p_{\theta_t^{(i)}}) = \|m^{(i)} - m_t^{(i)}\|^2 + \mathcal{B}^2(\Sigma^{(i)}, \Sigma_t^{(i)})\,.$$

The KL divergence is now written

$$\mathsf{KL}\Big(\frac{1}{N}\sum_{i=1}^{N}p_{\theta^{(i)}} \,\Big\|\, \pi\Big) = \frac{1}{N}\sum_{i=1}^{N}\int p_{\theta^{(i)}}\ln p - \frac{1}{N}\sum_{i=1}^{N}\int p_{\theta^{(i)}}\ln \pi\,.$$

For $k \in [N]$, the derivative of this divergence with respect to $m^{(k)}$ gives:

$$\nabla_{m^{(k)}}\mathsf{KL}(p\,\|\,\pi) = \int \frac{1}{N}\nabla_{m^{(k)}}p_{\theta^{(k)}}\ln p + \int p\,\nabla_{m^{(k)}}\ln p - \int \frac{1}{N}\nabla_{m^{(k)}}p_{\theta^{(k)}}\ln \pi$$

$$= \int \frac{1}{N}p_{\theta^{(k)}}\nabla_x \ln \frac{p}{\pi}\,,$$

where we have used the same integration by parts as in the Section A.1, i.e., $\int p_{\theta^{(k)}}\nabla_{m^{(k)}}\ln p = \int p_{\theta^{(k)}}\nabla_x \ln p$, and the Fisher score property $\int p\,\nabla_{m^{(k)}}\ln p = 0$. Mimicking Section A.1, see (15), we obtain in the limit $h \searrow 0$

$$\dot{m}^{(k)} = -\mathbb{E}_{p_{\theta^{(k)}}}\Big[\nabla_x \ln \frac{p}{\pi}\Big]\,,$$

which is the desired equation (11).

The derivative of the KL divergence with respect to $\Sigma^{(k)}$ gives:

$$\nabla_{\Sigma^{(k)}}\mathsf{KL}(p\,\|\,\pi) = \int \frac{1}{N}\nabla_{\Sigma^{(k)}}p_{\theta^{(k)}}\ln p + \int p\,\nabla_{\Sigma^{(k)}}\ln p - \int \frac{1}{N}\nabla_{\Sigma^{(k)}}p_{\theta^{(k)}}\ln \pi$$

$$= \frac{1}{N}\Big(\frac{1}{2}\int p_{\theta^{(k)}}\nabla_x^2 \ln p - \frac{1}{2}\int p_{\theta^{(k)}}\nabla_x^2 \ln \pi\Big) = \frac{1}{2N}\mathbb{E}_{p_{\theta^{(k)}}}\Big[\nabla_x^2 \ln \frac{p}{\pi}\Big]\,,$$

where we have used a double integration by parts $\int \nabla_{\Sigma^{(k)}}p_{\theta^{(k)}}\ln p = \frac{1}{2}\int p_{\theta^{(k)}}\nabla_x^2 \ln p$ as in Section A.1 and the Fisher score property $\int p\,\nabla_{\Sigma^{(k)}}\ln p = 0$.

Using the Bures derivative, the critical points of the variational loss with respect to $\Sigma_k$ satisfy:

$$\frac{1}{2Nh}\big(I - T^{\Sigma^{(k)},\Sigma_t^{(k)}}\big) + \frac{1}{2N}\mathbb{E}_{p_{\theta^{(k)}}}\Big[\nabla_x^2 \ln \frac{p}{\pi}\Big] = 0\,.$$

Multiplying on the left and on the right by $\Sigma^{(k)}$ and taking the average as in Section A.1, we find:

$$\frac{1}{h}\Big(\Sigma^{(k)} - \frac{1}{2}\big(\Sigma^{(k)}T^{\Sigma^{(k)},\Sigma_t^{(k)}} + T^{\Sigma^{(k)},\Sigma_t^{(k)}}\Sigma^{(k)}\big)\Big)$$

$$= -\frac{1}{2}\Big(\mathbb{E}_{p_{\theta^{(k)}}}\Big[\nabla_x^2 \ln \frac{p}{\pi}\Big]\Sigma^{(k)} + \Sigma^{(k)}\mathbb{E}_{p_{\theta^{(k)}}}\Big[\nabla_x^2 \ln \frac{p}{\pi}\Big]\Big)\,.$$

We can now use the first-order approximation $\frac{1}{2}\big(\Sigma^{(k)}T^{\Sigma^{(k)},\Sigma_t^{(k)}} + T^{\Sigma^{(k)},\Sigma_t^{(k)}}\Sigma^{(k)}\big) \approx \Sigma^{(k)} - \frac{h}{2}\dot{\Sigma}^{(k)}$ shown in Section A.1 to obtain:

$$\dot{\Sigma}^{(k)} = -\mathbb{E}_{p_{\theta^{(k)}}}\Big[\nabla_x^2 \ln \frac{p}{\pi}\Big]\Sigma^{(k)} - \Sigma^{(k)}\mathbb{E}_{p_{\theta^{(k)}}}\Big[\nabla_x^2 \ln \frac{p}{\pi}\Big]\,.$$

This yields the desired ODE (12), which can be rewritten in a Hessian-free form:

$$\dot{\Sigma}^{(k)} = -\mathbb{E}_{p_{\theta^{(k)}}}\Big[\nabla_x \ln \frac{p}{\pi} \otimes (x - m_k)\Big] - \mathbb{E}_{p_{\theta^{(k)}}}\Big[(x - m_k) \otimes \nabla_x \ln \frac{p}{\pi}\Big]\,.$$

# B Background on Otto calculus

## B.1 Overview and history

Historically, the connection between dissipative evolution equations and the theory of gradient flows on the Wasserstein space was discovered in Otto [1998]. Subsequently, this link was further developed and strengthened in the seminal works Jordan et al. [1998], Otto [2001]. Although the paper Jordan et al. [1998] chronologically precedes Otto [2001], the intuition of the former is based heavily on the work of Otto in the latter paper, in which he develops the formal[2] rules governing the calculus which now bears his name.

---

[2]Here, "formal" is not a synonym for "rigorous".

*Otto calculus* endows the space $\mathcal{P}_2(\mathbb{R}^d)$ of probability measures over $\mathbb{R}^d$ with finite second moment with a formal Riemannian structure inspired by fluid dynamics. To describe the idea, suppose that $(\mu_t)_{t \geq 0}$ is a curve of probability measures, with $\mu_t$ representing the fluid density at time $t$. Also, let $(v_t)_{t \geq 0}$ denote the velocity vector fields governing the dynamics of the particles; this means that the trajectory $t \mapsto x_t$ of an individual particle evolves according to the ODE

$$\dot{x}_t = v_t(x_t) \,. \tag{24}$$

In probabilistic language, if $x_0$ is a random variable drawn from the density $\mu_0$ and it evolves according to (24), then $x_t \sim \mu_t$ for all $t \geq 0$. From this, we can derive a partial differential equation (PDE) governing the evolution of $(\mu_t)_{t \geq 0}$ as follows: fix a test function $\varphi : \mathbb{R}^d \to \mathbb{R}$ (which is bounded, smooth, etc.). Formally, if the integration by parts is justified, then

$$\int \varphi \, \partial_t \mu_t = \partial_t \int \varphi \, \mathrm{d}\mu_t = \partial_t \, \mathbb{E} \, \varphi(x_t) = \int \langle \nabla \varphi, v_t \rangle \, \mathrm{d}\mu_t = - \int \varphi \, \mathrm{div}(\mu_t v_t)$$

from which we deduce the *continuity equation* of fluid dynamics:

$$\partial_t \mu_t + \mathrm{div}(\mu_t v_t) = 0 \,. \tag{25}$$

Conversely, if $(\mu_t)_{t \geq 0}$ is a sufficiently nice curve, then it is always possible to find a family of vector fields $(v_t)_{t \geq 0}$ such that the equation (25) holds, i.e., we can interpret $(\mu_t)_{t \geq 0}$ as the evolution of a fluid density. However, the choice of vector fields is not unique, since we may always replace $v_t$ with another vector field $\tilde{v}_t$ such that $\mathrm{div}(\mu_t \, (v_t - \tilde{v}_t)) = 0$. This motivates the search for a *distinguished* choice of vector fields to describe the evolution of the curve of measures.

To do so, we pick $v_t$ to minimize the *kinetic energy*,

$$v_t = \arg\min \left\{ \int \|w_t\|^2 \, \mathrm{d}\mu_t \;\middle|\; w_t : \mathbb{R}^d \to \mathbb{R} \;\text{ satisfies }\; \mathrm{div}(\mu_t w_t) = -\partial_t \mu_t \right\}.$$

If $\mu_t$ is regular (admits a density w.r.t. Lebesgue measure), then the minimum is attained at a gradient vector field: $v_t = \nabla \psi_t$ for a function $\psi_t : \mathbb{R}^d \to \mathbb{R}$. We are led to define the tangent space

$$T_\mu \mathcal{P}_2(\mathbb{R}^d) = \{\nabla \psi \mid \psi : \mathbb{R}^d \to \mathbb{R}\}$$

and endow it with the inner product

$$\langle v, w \rangle_\mu = \int \langle v, w \rangle \, \mathrm{d}\mu \,.$$

This yields a formal Riemannian structure on $\mathcal{P}_2(\mathbb{R}^d)$. Moreover, the choice of picking the vector field with minimal kinetic energy is closely related to the idea of optimal transport of mass [see Villani, 2003], and in fact Benamou and Brenier [1999] showed that

$$W_2^2(\mu_0, \mu_1) = \inf \left\{ \int \|v_t\|_{\mu_t}^2 \, \mathrm{d}t \;\middle|\; (\mu_t, v_t)_{t \in [0,1]} \;\text{ solves the continuity equation (25)} \right\}. \tag{26}$$

From the lens of Riemannian geometry, this says that the notion of distance induced by the Riemannian structure is precisely the quadratic Wasserstein distance, and hence we refer to the space $\mathcal{P}_2(\mathbb{R}^d)$ equipped with this Riemannian structure as the *Wasserstein space*.

This formal picture already allows one to compute gradients of functionals defined over $\mathcal{P}_2(\mathbb{R}^d)$ and hence to consider gradient flows, as well as to derive criteria which imply quantitative rates of convergence for these flows. However, it is a considerable technical undertaking to make the preceding formal considerations fully rigorous, and this was only accomplished later in the comprehensive monograph Ambrosio et al. [2008]. Instead, in Jordan et al. [1998], the authors sidestep this difficulty by considering an implicit time-discretization scheme which only requires the metric structure of $(\mathcal{P}_2(\mathbb{R}^d), W_2)$. For a step size $h > 0$, define the discrete updates

$$\mu_{h,k+1} := \arg\min_{\mu \in \mathcal{P}_2(\mathbb{R}^d)} \left\{ \mathcal{F}(\mu) + \frac{1}{2h} \, W_2^2(\mu, \mu_{h,k}) \right\}, \tag{27}$$

where $\mathcal{F} : \mathcal{P}_2(\mathbb{R}^d) \to \mathbb{R} \cup \{\infty\}$ is the functional of interest defined over the Wasserstein space. Note that in optimization, this is known as the "proximal point method" for minimizing $\mathcal{F}$.

As $h \searrow 0$, one hopes that we have convergence $\mu_{h,\lfloor t/h \rfloor} \to \mu_t$ in a suitable sense, and then the limiting curve $(\mu_t)_{t \geq 0}$ can be interpreted as the Wasserstein gradient flow of $\mathcal{F}$. This is indeed what Jordan et al. [1998] showed in a particular, but important case. Namely, if $\pi \propto \exp(-V)$ is a density on $\mathbb{R}^d$ obeying mild regularity conditions, and we take the functional to be the KL divergence, $\mathcal{F}(\mu) = \mathsf{KL}(\mu \parallel \pi)$, then the sequence of discrete approximations converges to the solution of the Fokker–Planck equation

$$\partial_t \mu_t = \mathrm{div}\big(\mu_t \nabla \ln \frac{\mu_t}{\pi}\big). \tag{28}$$

It is well-known that the Fokker–Planck equation governs the evolution of the marginal law of the Langevin diffusion

$$\mathrm{d}X_t = -\nabla V(X_t)\,\mathrm{d}t + \sqrt{2}\,\mathrm{d}B_t \,,$$

where $(B_t)_{t \geq 0}$ is a standard Brownian motion on $\mathbb{R}^d$. Hence, this celebrated result says that the Langevin diffusion can be interpreted as the Wasserstein gradient flow of the KL divergence. The implicit discretization (27) is now commonly known as the "JKO scheme" after the authors Jordan, Kinderlehrer, and Otto.

Although the Wasserstein space is not truly a Riemannian manifold, many of the formal calculations of Otto [2001] can now be justified rigorously, under appropriate technical conditions, due to the extensive theory developed in Ambrosio et al. [2008], Villani [2009]. This perspective leads to intuitive derivations of gradient flows, as explained in Section C, and much more.

## B.2 Geometry of the Wasserstein space

In this section, we provide further details about the geometry of $(\mathcal{P}_2(\mathbb{R}^d), W_2)$.

Let $\mu_0, \mu_1 \in \mathcal{P}_2(\mathbb{R}^d)$, and for simplicity assume that $\mu_0$ admits a density with respect to Lebesgue measure. Then, Brenier's theorem [Villani, 2003, Theorem 2.12] says that there exists a proper, convex, lower semicontinuous $\varphi : \mathbb{R}^d \to \mathbb{R} \cup \{\infty\}$ such that $\nabla\varphi$ solves the optimal transport problem from $\mu_0$ to $\mu_1$: namely, $(\nabla\varphi)_{\#}\mu_0 = \mu_1$ and $W_2^2(\mu_0, \mu_1) = \int \|\nabla\varphi(x) - x\|^2\,\mathrm{d}\mu_0(x)$. We refer to $\nabla\varphi$ as the *optimal transport map* from $\mu_0$ to $\mu_1$.

The (unique) constant-speed geodesic $(\mu_t)_{t \in [0,1]}$ joining $\mu_0$ to $\mu_1$ is then described via

$$\mu_t = (\nabla\varphi_t)_{\#}\mu_0 \,, \qquad \nabla\varphi_t := (1-t)\,\mathrm{id} + t\,\nabla\varphi \,. \tag{29}$$

In view of the fluid dynamical perspective, the constant-speed geodesics in the Wasserstein space correspond to particle trajectories $t \mapsto x_t$ which are straight lines traversed at constant speed: indeed, $x_t = \nabla\varphi_t(x_0) = (1-t)\,x_0 + t\,\nabla\varphi(x_0)$. Since $\dot{x}_t = \nabla\varphi(x_0) - x_0 = (\nabla\varphi - \mathrm{id}) \circ (\nabla\varphi_t)^{-1}(x_t)$, then along the geodesic we see that $(\mu_t, v_t)_{t \in [0,1]}$ solves the continuity equation (25), where the vector field is $v_t = (\nabla\varphi - \mathrm{id}) \circ (\nabla\varphi_t)^{-1}$. This solution achieves the minimum in (26).

Recall that on a Riemannian manifold $\mathcal{M}$, the Riemannian exponential map at $p$ is defined on a subset of the tangent space $T_p\mathcal{M}$, and it maps $v$ to the endpoint of the constant-speed geodesic at time 1 which emanates from $p$ with velocity $v$ (at time 0). The Riemannian logarithmic map $\log_p$ is the inverse mapping: it maps an element $q \in \mathcal{M}$ to the element $v \in T_p\mathcal{M}$ such that the constant-speed geodesic joining $p$ to $q$ in one unit of time has velocity $v$ at time 0. In the previous paragraph, we have identified the logarithmic map: $\log_\mu \nu = \nabla\varphi_{\mu \to \nu} - \mathrm{id}$, where $\nabla\varphi_{\mu \to \nu}$ is the optimal transport map from $\mu$ to $\nu$. Thus, the Riemannian exponential map is $\exp_\mu v = (\mathrm{id} + v)_{\#}\mu$.

## B.3 The Bures–Wasserstein space

The space of non-degenerate Gaussian distributions equipped with the $W_2$ metric is known as the Bures–Wasserstein space, after Bures [1969]. We denote this space as $\mathsf{BW}(\mathbb{R}^d)$.

Given $m \in \mathbb{R}^d$ and $\Sigma \succ 0$, we denote by $p_{m,\Sigma}$ the Gaussian on $\mathbb{R}^d$ with mean $m$ and covariance $\Sigma$. Conversely, for a non-degenerate Gaussian $p$ we write $(m_p, \Sigma_p)$ for its mean and covariance. Via this correspondence, we can therefore identify the space of non-degenerate Gaussians with the manifold $\mathbb{R}^d \times \mathbf{S}_{++}^d$, where $\mathbf{S}_{++}^d$ denotes the cone of positive definite matrices. Abusing notation, we will do so whenever there is no danger of confusion.

Suppose that $p_{m_0, \Sigma_0}, p_{m_1, \Sigma_1} \in \mathsf{BW}(\mathbb{R}^d)$. Then, the optimal transport map from $p_0 := p_{m_0, \Sigma_0}$ to $p_1 := p_{m_1, \Sigma_1}$ is

$$\nabla \varphi(x) = m_1 + \Sigma_0^{-1/2} \left( \Sigma_0^{1/2} \Sigma_1 \Sigma_0^{1/2} \right)^{1/2} \Sigma_0^{-1/2} \left( x - m_0 \right).$$

Observe that $\nabla \varphi$ is an affine map. Since the pushforward of a Gaussian via an affine map is also Gaussian, it follows from (29) that the constant speed geodesic $(p_t)_{t \in [0,1]}$ joining $p_0$ to $p_1$ also lies in $\mathsf{BW}(\mathbb{R}^d)$. In other words, $\mathsf{BW}(\mathbb{R}^d)$ is a *geodesically convex* subset of $\mathcal{P}_2(\mathbb{R}^d)$.

The tangent vector to the geodesic at time 0 is always an affine map of the form $x \mapsto a + S \left( x - m_{p_0} \right)$, where $a \in \mathbb{R}^d$ and $S$ is a symmetric matrix. The tangent space is

$$T_p \mathsf{BW}(\mathbb{R}^d) = \{ x \mapsto a + S \left( x - m_p \right) \mid a \in \mathbb{R}^d, \ S \in \mathbf{S}^d \},$$

which can therefore be identified with pairs $(a, S) \in \mathbb{R}^d \times \mathbf{S}^d$. With this abuse of notation, if $(a, S), (a', S') \in T_p \mathsf{BW}(\mathbb{R}^d)$, then

$$\langle (a, S), (a', S') \rangle_p = \int \langle a + S \left( x - m_p \right), a' + S' \left( x - m_p \right) \rangle \, \mathrm{d}p(x) = \langle a, a' \rangle + \langle S, \Sigma_p S' \rangle. \quad (30)$$

Specializing the notions from the previous section, we obtain

$$\log_p(q) = \left( m_q - m_p, \ \Sigma_p^{-1/2} \left( \Sigma_p^{1/2} \Sigma_q \Sigma_p^{1/2} \right)^{1/2} \Sigma_p^{-1/2} - I \right),$$
$$\exp_p(a, S) = \left( m_p + a + (S + I) \left( \cdot - m_p \right) \right)_{\#} p = \mathcal{N} \left( m_p + a, \ (S + I) \Sigma_p (S + I) \right).$$

Here, $\exp_p(a, S)$ is defined if $S \succ -I$.

This definition of the tangent space is consistent with the Wasserstein space, in that we have the inclusion $T_p \mathsf{BW}(\mathbb{R}^d) \hookrightarrow T_p \mathcal{P}_2(\mathbb{R}^d)$, but the abuse of notation $T_p \mathsf{BW}(\mathbb{R}^d) = \mathbb{R}^d \times \mathbf{S}^d$ can sometimes cause confusion. Indeed, if $(p_t = p_{m_t, \Sigma_t})_{t \in [0,1]}$ is a constant-speed geodesic in $\mathsf{BW}(\mathbb{R}^d)$, and the tangent vector at time 0 is $(a, S)$, then

$$p_t = \exp_{p_0} \left( t \, (a, S) \right) = \mathcal{N} \left( m_p + ta, \ (tS + I) \Sigma_p (tS + I) \right).$$

In particular, $\Sigma_t \neq \Sigma_0 + t \left( S - I \right)$, and

$$\dot{m}_0 = a, \quad (31)$$
$$\dot{\Sigma}_0 = S\Sigma_0 + \Sigma_0 S. \quad (32)$$

Although we derived the equations (31) and (32) for geodesic curves, they also hold for any curve $(p_t)_{t \geq 0}$ with tangent vector equal to $(a, S)$ at time 0. Using this, we can derive an expression for the Bures–Wasserstein gradient $\nabla_{\mathsf{BW}} f$ of a function $f : \mathbb{R}^d \times \mathbf{S}_{++}^d \to \mathbb{R}$. By definition, this satisfies, for any curve $(m_t, \Sigma_t)_{t \geq 0}$ with tangent vector $(a, S)$ at time 0,

$$\langle \nabla_{\mathsf{BW}} f(m_0, \Sigma_0), (a, S) \rangle_{p_{m_0, \Sigma_0}} = \partial_t \big|_{t=0} f(m_t, \Sigma_t).$$

Write $(\bar{a}, \bar{S}) = \nabla_{\mathsf{BW}} f(m_0, \Sigma_0)$. Then, we want

$$\langle \bar{a}, a \rangle + \langle \bar{S}, \Sigma_0 S \rangle = \langle \nabla_m f(m_0, \Sigma_0), \dot{m}_0 \rangle + \langle \nabla_\Sigma f(m_0, \Sigma_0), \dot{\Sigma}_0 \rangle$$
$$= \langle \nabla_m f(m_0, \Sigma_0), a \rangle + 2 \langle \nabla_\Sigma f(m_0, \Sigma_0), \Sigma_0 S \rangle,$$

where $\nabla_m, \nabla_\Sigma$ denote the usual Euclidean gradients. Hence, by identification, we conclude that the Bures–Wasserstein gradient of $f$ is related to the Euclidean gradient of $f$ via

$$\nabla_{\mathsf{BW}} f(m, \Sigma) = \left( \nabla_m f(m, \Sigma), \ 2 \nabla_\Sigma f(m, \Sigma) \right). \quad (33)$$

See [Altschuler et al., 2022, Appendix A] for further discussion.

## B.4 Evolution of the mean and covariance along the Fokker–Planck equation

It is known that the Wasserstein gradient of $\mathcal{F} := \mathsf{KL}(\cdot \parallel \pi)$ is

$$\nabla_{W_2} \mathcal{F}(\mu) = \nabla \ln \frac{\mu}{\pi}\,. \tag{34}$$

[See, e.g., Ambrosio et al., 2008, Theorem 10.4.13.] Also, as shown by Jordan et al. [1998], the Langevin diffusion is the gradient flow of $\mathsf{KL}(\cdot \parallel \pi)$. In Otto calculus, this means that the law $(\pi_t)_{t \geq 0}$ of the Langevin diffusion obeys the continuity equation (25) with velocity vector field $v_t = -\nabla_{W_2} \mathcal{F}(\pi_t) = -\nabla \ln(\pi_t/\pi)$, which is consistent with the Fokker–Planck equation (28).

According to the particle interpretation (24) of dynamics in the Wasserstein space, if $x_0 \sim \pi_0$ and

$$\dot{x}_t = v_t(x_t) = -\nabla \ln \frac{\pi_t}{\pi}(x_t)\,,$$

then $x_t \sim \pi_t$. Note that $(x_t)_{t \geq 0}$ is *not* the Langevin diffusion (1) as it is the solution to a deterministic ODE (albeit with random initial condition), but the marginal law of $(x_t)_{t \geq 0}$ agrees with that of the Langevin diffusion. This provides a convenient tool for calculating the evolution of the mean and covariance along the Fokker–Planck equation, as we now demonstrate.

The evolution of the mean is

$$\dot{m}_t = \partial_t \, \mathbb{E} \, x_t = \mathbb{E} \, \dot{x}_t = -\, \mathbb{E} \, \nabla \ln \frac{\pi_t}{\pi}(x_t)\,.$$

Since $\mathbb{E} \, \nabla \ln \pi_t(x_t) = 0$ (which is verified via integration by parts), and $\pi \propto e^{-V}$, this can also be written as

$$\dot{m}_t = -\, \mathbb{E}_{\pi_t} \, \nabla V\,.$$

Next, for the evolution of the covariance,

$$\partial_t \, \mathbb{E}(x_t \otimes x_t) = \mathbb{E}(x_t \otimes \dot{x}_t + \dot{x}_t \otimes x_t) = -\, \mathbb{E}\big(x_t \otimes \nabla \ln \frac{\pi_t}{\pi}(x_t) + \nabla \ln \frac{\pi_t}{\pi}(x_t) \otimes x_t\big)$$

$$\partial_t \, \mathbb{E}(x_t) \otimes \mathbb{E}(x_t) = m_t \otimes \mathbb{E}(\dot{x}_t) + \mathbb{E}(\dot{x}_t) \otimes m_t = -\, \mathbb{E}\big(m_t \otimes \nabla \ln \frac{\pi_t}{\pi}(x_t) + \nabla \ln \frac{\pi_t}{\pi}(x_t) \otimes m_t\big)$$

which yields

$$\dot{\Sigma}_t = -\, \mathbb{E}\big((x_t - m_t) \otimes \nabla \ln \frac{\pi_t}{\pi}(x_t) + \nabla \ln \frac{\pi_t}{\pi}(x_t) \otimes (x_t - m_t)\big)\,.$$

Integration by parts yields

$$\int (\bullet - m_t) \otimes \nabla \ln \pi_t \, \mathrm{d}\pi_t + \int \nabla \ln \pi_t \otimes (\bullet - m_t) \, \mathrm{d}\pi_t$$

$$= \int (\bullet - m_t) \otimes \nabla \pi_t + \int \nabla \pi_t \otimes (\bullet - m_t) = -2I\,.$$

Hence,

$$\dot{\Sigma}_t = 2I - \mathbb{E}_{\pi_t}[\nabla V \otimes (\bullet - m_t) + (\bullet - m_t) \otimes \nabla V]\,.$$

This verifies equation (3). The equations in this section can also be derived using Itô calculus.

# C  Proofs via Otto calculus

Our aim in this section is to derive the Wasserstein gradient flow of the KL divergence $\mathsf{KL}(\cdot \parallel \pi)$ *constrained* to lie in the Bures–Wasserstein space of non-degenerate Gaussian measures.

Since the Bures–Wasserstein space can be formally viewed as a submanifold of the Wasserstein space, it leads to two natural approaches for computing the constrained gradient flow. In the first approach, we take the Wasserstein gradient of $\mathsf{KL}(\cdot \parallel \pi)$ and we compute the orthogonal projection onto the tangent space of the Bures–Wasserstein space. In the second approach, we note that the geometry of the Bures–Wasserstein space has been studied in its own right [see, e.g., Bhatia et al., 2019] and in particular, the explicit expression (33) for the Bures–Wasserstein gradient is known. We can therefore view $\mathsf{KL}(\cdot \parallel \pi)$ as a functional over $\mathsf{BW}(\mathbb{R}^d)$ and compute its gradient directly using (33).

## C.1 Orthogonal projection approach

First, we justify why computing the orthogonal projection of the $\mathcal{P}_2(\mathbb{R}^d)$ gradient gives the same result as computing the intrinsic gradient on $\mathsf{BW}(\mathbb{R}^d)$. Let $\mathcal{F}$ be any functional on $\mathcal{P}_2(\mathbb{R}^d)$. By definition, the Bures–Wasserstein gradient $\nabla_{\mathsf{BW}}\mathcal{F}$ satisfies

$$\partial_t \mathcal{F}(p_t) = \langle \nabla_{\mathsf{BW}}\mathcal{F}(p_t), v_t \rangle_{p_t} \tag{35}$$

for any curve $(p_t)_{t\in\mathbb{R}}$ in $\mathsf{BW}(\mathbb{R}^d)$ with tangent vectors $(v_t)_{t\in\mathbb{R}}$. Here, $\nabla_{\mathsf{BW}}\mathcal{F}(p_t) \in T_{p_t}\mathsf{BW}(\mathbb{R}^d)$. On the other hand, since $(p_t)_{t\geq 0}$ is also a curve in $\mathcal{P}_2(\mathbb{R}^d)$ and the Riemannian structure of $\mathsf{BW}(\mathbb{R}^d)$ is consistent with that of $\mathcal{P}_2(\mathbb{R}^d)$, the definition of the gradient in $\mathcal{P}_2(\mathbb{R}^d)$ yields

$$\partial_t \mathcal{F}(p_t) = \langle \nabla_{W_2}\mathcal{F}(p_t), v_t \rangle_{p_t}\,.$$

Note that the orthogonal projection

$$\mathrm{proj}_{T_{p_t}\mathsf{BW}(\mathbb{R}^d)} \nabla_{W_2}\mathcal{F}(p_t) = \underset{w \in T_{p_t}\mathsf{BW}(\mathbb{R}^d)}{\arg\min} \|w - \nabla_{W_2}\mathcal{F}(p_t)\|_{p_t}^2$$

is characterized as the unique element of $T_{p_t}\mathsf{BW}(\mathbb{R}^d)$ satisfying

$$\langle \mathrm{proj}_{T_{p_t}\mathsf{BW}(\mathbb{R}^d)} \nabla_{W_2}\mathcal{F}(p_t), v \rangle_{p_t} = \langle \nabla_{W_2}\mathcal{F}(p_t), v \rangle_{p_t}$$

for all $v \in T_{p_t}\mathsf{BW}(\mathbb{R}^d)$. Thus, (35) holds with

$$\nabla_{\mathsf{BW}}\mathcal{F}(p) = \mathrm{proj}_{T_p\mathsf{BW}(\mathbb{R}^d)} \nabla_{W_2}\mathcal{F}(p)\,.$$

This argument clearly works for arbitrary Riemannian submanifolds.

Next, we compute the projection of the $\mathcal{P}_2(\mathbb{R}^d)$ gradient of the KL divergence.

Using the formula (34) for the $\mathcal{P}_2(\mathbb{R}^d)$ gradient of the KL divergence and the description of the tangent space to $\mathsf{BW}(\mathbb{R}^d)$ in Section B.3 and (30), the projected gradient $(\bar{a}, \bar{S}) \in \mathbb{R}^d \times \mathbf{S}^d$ is such that for all $(a, S) \in \mathbb{R}^d \times \mathbf{S}^d$,

$$\int \Big\langle \nabla \ln \frac{p}{\pi}(x), a + S\,(x - m_p) \Big\rangle \, \mathrm{d}p(x) = \langle (\bar{a}, \bar{S}), (a, S) \rangle_p = \langle \bar{a}, a \rangle + \langle \bar{S}, \Sigma_p S \rangle\,.$$

Using $\nabla p(x) = -\Sigma_p^{-1}\,(x - m_p)\,p(x)$ and integration by parts,

$$\int \Big\langle \nabla \ln \frac{p}{\pi}(x), a + S\,(x - m_p) \Big\rangle \, \mathrm{d}p(x)$$

$$= \Big\langle \mathbb{E}_p \nabla \ln \frac{p}{\pi}, a \Big\rangle + \int \Big\langle \Sigma_p S \nabla \ln \frac{p}{\pi}(x), \Sigma_p^{-1}\,(x - m_p) \Big\rangle \, \mathrm{d}p(x)$$

$$= \Big\langle \mathbb{E}_p \nabla \ln \frac{p}{\pi}, a \Big\rangle - \int \Big\langle \Sigma_p S \nabla \ln \frac{p}{\pi}(x), \nabla p(x) \Big\rangle \, \mathrm{d}x$$

$$= \Big\langle \mathbb{E}_p \nabla \ln \frac{p}{\pi}, a \Big\rangle + \int \mathrm{div}\Big(\Sigma_p S \nabla \ln \frac{p}{\pi}\Big)(x) \, \mathrm{d}p(x)$$

$$= \Big\langle \mathbb{E}_p \nabla \ln \frac{p}{\pi}, a \Big\rangle + \Big\langle \mathbb{E}_p \nabla^2 \ln \frac{p}{\pi}, \Sigma_p S \Big\rangle\,.$$

Hence,

$$(\bar{a}, \bar{S}) = \Big(\mathbb{E}_p \nabla \ln \frac{p}{\pi},\ \mathbb{E}_p \nabla^2 \ln \frac{p}{\pi}\Big)\,. \tag{36}$$

Using the fact that $\mathbb{E}_p \nabla \ln p = 0$, this can also be written

$$(\bar{a}, \bar{S}) = \big(\mathbb{E}_p \nabla V,\ \mathbb{E}_p \nabla^2 V - \Sigma_p^{-1}\big)$$

which corresponds to the affine map

$$x \mapsto \mathbb{E}_p \nabla V + \big(\mathbb{E}_p \nabla^2 V - \Sigma_p^{-1}\big)\,(x - m_p)\,. \tag{37}$$

If $(p_t = p_{m_t, \Sigma_t})_{t \geq 0}$ evolves according to the constrained gradient flow, then using the expression for the projected Wasserstein gradient together with (31) and (32),

$$\begin{aligned}
\dot{m}_t &= -\mathbb{E}_{p_t} \nabla V \,, \\
\dot{\Sigma}_t &= 2I - \Sigma_t \, \mathbb{E}_{p_t} \nabla^2 V - \mathbb{E}_{p_t} \nabla^2 V \, \Sigma_t \,.
\end{aligned}$$

The sign in the above equations comes from the fact that we perform steepest *descent* in Bures–Wasserstein descent, i.e., the tangent vector to the curve at time $t$ is $-\operatorname{proj}_{T_{p_t} \mathsf{BW}(\mathbb{R}^d)} \nabla_{W_2} \mathcal{F}(p_t)$.

The system of equations we have derived here differs from the system (4), but we can check that they agree using integration by parts. Indeed,

$$\begin{aligned}
\dot{\Sigma}_t &= 2I - \Sigma_t \int \nabla^2 V \, \mathrm{d}p_t - \int \nabla^2 V \, \mathrm{d}p_t = 2I + \Sigma_t \int \nabla p_t \otimes \nabla V + \int \nabla V \otimes \nabla p_t \, \Sigma_t \\
&= 2I + \Sigma_t \int \nabla \ln p_t \otimes \nabla V \, \mathrm{d}p_t + \int \nabla V \otimes \nabla \ln p_t \, \mathrm{d}p_t \, \Sigma_t \\
&= 2I - \mathbb{E}_{p_t}[(\bullet - m_t) \otimes \nabla V + \nabla V \otimes (\bullet - m_t)] \,.
\end{aligned}$$

### C.2 Alternate proof using direct Bures–Wasserstein calculation

In the second approach, we view $\mathcal{F}$ as a functional on the Bures–Wasserstein space. Explicitly,

$$\mathcal{F}(m, \Sigma) = \int p_{m, \Sigma} \ln \frac{p_{m, \Sigma}}{\pi} \,.$$

Using (33),

$$\begin{aligned}
\nabla_{\mathsf{BW}} \mathcal{F}(m, \Sigma) &= \bigl(\nabla_m \mathcal{F}(m, \Sigma), \, 2 \, \nabla_\Sigma \mathcal{F}(m, \Sigma)\bigr) \\
&= \left( \int \nabla_m p_{m, \Sigma} \ln \frac{p_{m, \Sigma}}{\pi}, \, 2 \int \nabla_\Sigma p_{m, \Sigma} \ln \frac{p_{m, \Sigma}}{\pi} \right) . \tag{38}
\end{aligned}$$

Furthermore, using the identities

$$\nabla_m p_{m, \Sigma}(x) = -\nabla_x p_{m, \Sigma}(x) \qquad \text{and} \qquad \nabla_\Sigma p_{m, \Sigma}(x) = \frac{1}{2} \, \nabla_x^2 p_{m, \Sigma}(x) \tag{39}$$

for the Gaussian distribution, integration by parts verifies that (38) agrees with (36).

## D   Proof of Corollary 1

Corollary 1 is a consequence of general and well-known principles for gradient flows. To emphasize this generality, we will consider an abstract $\alpha$-convex differentiable functional $\mathcal{F}$ defined over a geodesically convex subset of a Riemannian manifold; this ensures that the logarithmic map is well-defined in the following calculations. We assume that $\mathcal{F}$ is minimized at $p^\star$; by adding a constant to $\mathcal{F}$, we can assume $\inf \mathcal{F} = 0$. Let $\mathsf{d}$ denote the distance function on the manifold. If $(p_t)_{t \geq 0}$, $(q_t)_{t \geq 0}$ are two solutions to the gradient flow for $\mathcal{F}$, then

$$\partial_t \mathsf{d}^2(p_t, q_t) = 2 \left\langle \log_{p_t}(q_t), \nabla \mathcal{F}(p_t) \right\rangle_{p_t} + 2 \left\langle \log_{q_t}(p_t), \nabla \mathcal{F}(q_t) \right\rangle_{q_t} .$$

(The reader who is unfamiliar with Riemannian geometry should keep in mind that in Euclidean space, $\log_p(q) = q - p$.) Next, the $\alpha$-convexity of $\mathcal{F}$ implies

$$\begin{aligned}
\mathcal{F}(p_t) &\geq \mathcal{F}(q_t) + \langle \nabla \mathcal{F}(q_t), \log_{q_t}(p_t) \rangle_{q_t} + \frac{\alpha}{2} \, \mathsf{d}^2(p_t, q_t) \,, \\
\mathcal{F}(q_t) &\geq \mathcal{F}(p_t) + \langle \nabla \mathcal{F}(p_t), \log_{p_t}(q_t) \rangle_{p_t} + \frac{\alpha}{2} \, \mathsf{d}^2(p_t, q_t) \,.
\end{aligned}$$

Adding these equations and rearranging yields

$$\partial_t \mathsf{d}^2(p_t, q_t) \leq -2\alpha \, \mathsf{d}^2(p_t, q_t) \,.$$

By Grönwall's inequality, it implies

$$\mathsf{d}^2(p_t, q_t) \leq \exp(-2\alpha t) \, \mathsf{d}^2(p_0, q_0) \,.$$

This inequality has two consequences. First, for any $\alpha \in \mathbb{R}$, $p_0 = q_0$ implies $p_t = q_t$: the solution to the gradient flow is unique. Second, if $\alpha > 0$, then we can set $q_t = p^\star$ for all $t \geq 0$ to deduce exponential contraction of the gradient flow to the minimizer $p^\star$, which is the first statement of Corollary 1.

To obtain convergence in functional values, observe that by definition of the gradient flow, we have on the one hand that

$$\partial_t \mathcal{F}(p_t) = -\|\nabla \mathcal{F}(p_t)\|_{p_t}^2 . \tag{40}$$

On the other hand, if $\alpha > 0$, the convexity inequality and Young's inequality respectively, yield

$$0 = \mathcal{F}(p^\star) \geq \mathcal{F}(p) + \langle \nabla \mathcal{F}(p), \log_p(p^\star) \rangle_p + \frac{\alpha}{2} \, \mathsf{d}^2(p, p^\star) \tag{41}$$

$$\geq \mathcal{F}(p) - \frac{1}{2\alpha} \|\nabla \mathcal{F}(p)\|_p^2 - \frac{\alpha}{2} \underbrace{\|\log_p(p^\star)\|_p^2}_{=\mathsf{d}^2(p,p^\star)} + \frac{\alpha}{2} \, \mathsf{d}^2(p, p^\star)$$

and hence $\|\nabla \mathcal{F}(p)\|^2 \geq 2\alpha \, \mathcal{F}(p)$. Substituting this into (40) and applying Grönwall's inequality again, we deduce

$$\mathcal{F}(p_t) \leq \exp(-2\alpha t) \, \mathcal{F}(p_0) .$$

Finally, suppose $\alpha = 0$. We consider the Lyapunov functional

$$\mathcal{L}_t := t \, \mathcal{F}(p_t) + \frac{1}{2} \, \mathsf{d}^2(p_t, p^\star) .$$

Differentiating in time,

$$\partial_t \mathcal{L}_t = \mathcal{F}(p_t) - t \, \|\nabla \mathcal{F}(p_t)\|_{p_t}^2 + \langle \log_{p_t}(p^\star), \nabla \mathcal{F}(p_t) \rangle_{p_t} .$$

On the other hand, applying the convexity inequality in (41) with $\alpha = 0$ yields $\partial_t \mathcal{L}_t \leq 0$. Hence, $\mathcal{L}_t \leq \mathcal{L}_0$, and

$$\mathcal{F}(p_t) \leq \frac{\mathsf{d}^2(p_0, p^\star)}{2t} .$$

# E  Proof of Theorem 2

In this section, we use the Riemannian exponential and logarithmic maps, as discussed in Section B.3. Also, let $\mathcal{F} := \mathsf{KL}(\cdot \parallel \pi)$ denote the KL divergence.

For $\tau > 0$, the eigenvalue clipping operation is defined as

$$\operatorname{clip}^\tau : \qquad \Sigma = \sum_{i=1}^d \lambda_i u_i u_i^\mathsf{T} \quad \mapsto \quad \operatorname{clip}^\tau \Sigma := \sum_{i=1}^d (\lambda_i \wedge \tau) \, u_i u_i^\mathsf{T} . \tag{42}$$

In the proof of Theorem 1 in Section C, we showed that the Bures–Wasserstein gradient is

$$g_p := \nabla_{\mathsf{BW}} \mathcal{F}(p) = \left( \mathbb{E}_p \, \nabla V, \, \mathbb{E}_p \, \nabla^2 V - \Sigma^{-1} \right) \tag{43}$$

where $\Sigma$ is the covariance matrix of $p$. Here, the first component of the gradient governs the evolution of the mean, whereas the second component governs the evolution of the covariance; see Section B.3. We propose to estimate the gradient in (43) via a sample,

$$\hat{g}_p := \left( \nabla V(\hat{X}), \, \nabla^2 V(\hat{X}) - \Sigma^{-1} \right), \qquad \hat{X} \sim p .$$

By comparing Algorithm 1 and the definition of the exponential map in Section B.3, one can check that for $p_k^+ := p_{m_{k+1}, \Sigma_k^+}$ and[3] $h \leq 1$

$$p_k^+ = \exp_{p_k}(-h\hat{g}_k) ,$$

---

[3]This latter requirement is needed because $\mathsf{BW}(\mathbb{R}^d)$ has a finite injectivity radius.

where $\hat{g}_k \in T_{p_k}\mathsf{BW}(\mathbb{R}^d)$ is the stochastic gradient

$$\hat{g}_k(x) = \nabla V(\hat{X}_k) + (\nabla^2 V(\hat{X}_k) - \Sigma_k^{-1})(x - m_k).$$

Thus, aside from the eigenvalue clipping operation (which is harmless, due to Lemma 3 below), Algorithm 1 is exactly a stochastic gradient descent scheme on $\mathsf{BW}(\mathbb{R}^d)$. Note also that from the definition of the exponential map in Section B.2, the update can also be written at the particle level: if $X_k \sim p_k$ is independent of $\hat{g}_k$, then

$$X_k^+ := X_k - h\,\hat{g}_k(X_k) \sim p_k^+. \tag{44}$$

In the next lemma, we obtain a uniform control on the smallest eigenvalues of the covariance matrices of the iterates.

**Lemma 2.** *Assume that $0 \prec \alpha I \preceq \nabla^2 V \preceq I$ holds and $h \leq \alpha^2/60$. Also, in Algorithm 1, assume that $\Sigma_k \succeq \frac{\alpha}{9} I$. Then, $\Sigma_k^+ \succeq \frac{\alpha}{9} I$.*

*Proof.* Since the statement of the lemma only involves the covariance matrices, we can suppose that all of the mean vectors are zero.

The key is to write $\Sigma_k^+$ as a generalized Bures–Wasserstein barycenter at $\Sigma_k$ for an appropriate distribution. Recall that

$$\Sigma_k^+ = \left(I + h\,\Sigma_k^{-1} - h\,\nabla^2 V(\hat{X}_k)\right) \Sigma_k \left(I + h\,\Sigma_k^{-1} - h\,\nabla^2 V(\hat{X}_k)\right). \tag{45}$$

Note that $\Sigma_k^{-1}$ is the optimal transport map from the Gaussian $p_{0,\Sigma_k}$ to $p_{0,\Sigma_k^{-1}}$.[4] Hence,

$$h\,\Sigma_k^{-1} - h\,\nabla^2 V(\hat{X}_k) = h\,(\Sigma_k^{-1} - I) + h\,(I - \nabla^2 V(\hat{X}_k))$$
$$= h\log_{\Sigma_k}(\Sigma_k^{-1}) + h\log_{\Sigma_k}(\tilde{\Sigma})$$

where we defined the matrix $\tilde{\Sigma} = (2I - \nabla^2 V(\hat{X}_k))\,\Sigma_k\,(2I - \nabla^2 V(\hat{X}_k))$. To check that this is valid, we need $2I - \nabla^2 V(\hat{X}_k) \succeq 0$, i.e., $\nabla^2 V(\hat{X}_k) \preceq 2I$, which follows from $\nabla^2 V \preceq I$.

We have shown that

$$\Sigma_k^+ = \exp_{\Sigma_k}\left(\int \log_{\Sigma_k}(\Sigma)\,\mathrm{d}P(\Sigma)\right)$$

where

$$P = (1 - 2h)\,\delta_{\Sigma_k} + h\,\delta_{\Sigma_k^{-1}} + h\,\delta_{\tilde{\Sigma}} = (1 - 2h)\,\delta_{\Sigma_k} + 2h\left(\frac{1}{2}\,\delta_{\Sigma_k^{-1}} + \frac{1}{2}\,\delta_{\tilde{\Sigma}}\right).$$

This is precisely the definition of a generalized Bures–Wasserstein barycenter.

Next, suppose that $\Sigma_k \succeq \lambda I$ for some $\lambda > 0$. Since $\Sigma_k \preceq \alpha^{-1} I$, and $I \preceq 2I - \nabla^2 V(\hat{X}_k) \preceq 2I$,

$$\alpha I \preceq \Sigma_k^{-1} \preceq \frac{1}{\lambda} I, \qquad \text{and} \qquad \lambda I \preceq \tilde{\Sigma} \preceq \frac{4}{\alpha} I.$$

Then, [Altschuler et al., 2022, Theorem 1][5] implies the following. If we define the quantities

$$\lambda_- := \left(\frac{1}{2}\sqrt{\alpha} + \frac{1}{2}\sqrt{\lambda}\right)^2, \qquad \lambda_+ := \frac{1}{2}\frac{1}{\lambda} + \frac{1}{2}\frac{4}{\alpha},$$

then for step sizes $2h \leq \frac{\lambda_-}{2\lambda_+}$ and if $\Sigma_k \succeq \frac{\lambda_-}{4} I$, we also have $\Sigma_k^+ \succeq \frac{\lambda_-}{4} I$. To use this result, let us choose $\lambda$ such that $\frac{\lambda_-}{4} = \lambda$; it can be seen that this holds with $\lambda = \frac{\alpha}{9}$. Since $\lambda_+ = \frac{13}{2\alpha}$, the step size condition then translates into $h \leq \frac{2\alpha^2}{117}$, for which it suffices to have $h \leq \frac{\alpha^2}{60}$. $\qquad\square$

We also recall an important fact about the eigenvalue clipping operation.

---

[4]This observation was also used in the analysis of Bures–Wasserstein gradient descent for entropically regularized barycenters in Altschuler et al. [2022].

[5]See the latest revision.

**Lemma 3** ([Altschuler et al., 2022, Proposition 3]). *For any $m \in \mathbb{R}^d$, $\tau > 0$, and $\Sigma, \Sigma' \in \mathbf{S}_{++}^d$,*

$$W_2(p_{m,\mathrm{clip}^\tau \Sigma},\ p_{m,\mathrm{clip}^\tau \Sigma'}) \leq W_2(p_{m,\Sigma},\ p_{m,\Sigma'}).$$

We now turn towards the proof of Theorem 2. In the proof, we let $\mathscr{F}_k := \sigma(\hat{X}_0, \hat{X}_1, \hat{X}_2, \ldots, \hat{X}_{k-1})$ be the $\sigma$-algebra generated by the random samples up until iteration $k$.

*Proof of Theorem 2.* Conditioned on $\mathscr{F}_k$, and independently of $\hat{X}_k$, let $X_k \sim p_k$ and $Z \sim \hat{\pi}$ be optimally coupled; let $\bar{\mathbb{E}}$ denote the expectation taken w.r.t. $(X_k, Z)$. Using Lemma 3, the fact that $\hat{\Sigma} \preceq \frac{1}{\alpha} I$ (see discussion in Section 3.3), and (44), we have

$$\mathbb{E}[W_2^2(p_{k+1}, \hat{\pi}) \mid \mathscr{F}_k] \leq \mathbb{E}[W_2^2(p_k^+, \hat{\pi}) \mid \mathscr{F}_k]$$
$$\leq \mathbb{E}\big[\bar{\mathbb{E}}[\|X_k - h\,\hat{g}_k(X_k) - Z\|^2] \mid \mathscr{F}_k\big]$$
$$= \mathbb{E}\big[\bar{\mathbb{E}}[\|X_k - Z\|^2 - 2h\,\langle \hat{g}_k(X_k), X_k - Z\rangle + h^2\,\|\hat{g}_k(X_k)\|^2] \mid \mathscr{F}_k\big]$$
$$= W_2^2(p_k, \hat{\pi}) - 2h\,\bar{\mathbb{E}}\langle g_k(X_k), X_k - Z\rangle + h^2\,\mathbb{E}\big[\bar{\mathbb{E}}[\|\hat{g}_k(X_k)\|^2] \mid \mathscr{F}_k\big],$$

where we abbreviated $g_k := g_{p_k}$. From strong convexity of $\mathsf{KL}(\cdot \,\|\, \pi)$ on $\mathsf{BW}(\mathbb{R}^d)$ (Lemma 1),

$$\bar{\mathbb{E}}\langle g_k(X_k), X_k - Z\rangle \geq \mathsf{KL}(p_k \,\|\, \pi) - \mathsf{KL}(\hat{\pi} \,\|\, \pi) + \frac{\alpha}{2}\,W_2^2(p_k, \hat{\pi})$$
$$\geq \alpha\,W_2^2(p_k, \hat{\pi}).$$

Thus,

$$\mathbb{E}[W_2^2(p_{k+1}, \hat{\pi}) \mid \mathscr{F}_k] \leq (1 - 2\alpha h)\,W_2^2(p_k, \hat{\pi}) + h^2\,\underbrace{\mathbb{E}\big[\bar{\mathbb{E}}[\|\hat{g}_k(X_k)\|^2] \mid \mathscr{F}_k\big]}_{=:\mathsf{err}}.$$

It remains to bound the error term.

Recall that

$$\hat{g}_k(X_k) = (\nabla^2 V(\hat{X}_k) - \Sigma_k^{-1})\,(X_k - m_k) + \nabla V(\hat{X}_k).$$

We bound the terms one by one. First,

$$\bar{\mathbb{E}}[\|\Sigma_k^{-1}\,(X_k - m_k)\|^2] = \mathrm{tr}(\Sigma_k^{-1}) \leq \frac{9d}{\alpha}$$

where we used Lemma 2. Next, since $\nabla^2 V \preceq I$ by assumption,

$$\bar{\mathbb{E}}[\|\nabla^2 V(\hat{X}_k)\,(X_k - m_k)\|^2] \leq \bar{\mathbb{E}}[\|X_k - m_k\|^2] = \mathrm{tr}(\Sigma_k) \leq \frac{d}{\alpha}.$$

Lastly, let $\hat{Z} \sim \hat{\pi}$ be optimally coupled with $\hat{X}_k$. By the optimality condition for $\hat{\pi}$ (Section 3.3), we know that $\mathbb{E}\,\nabla V(\hat{Z}) = 0$. Applying the Poincaré inequality for $\hat{\pi}$ (which holds because $\hat{\pi}$ is strongly log-concave, see [Bakry et al., 2014, Theorem 4.8.4])

$$\bar{\mathbb{E}}[\|\nabla V(\hat{X}_k)\|^2] \leq 2\,\bar{\mathbb{E}}[\|\nabla V(\hat{Z})\|^2] + 2\,\bar{\mathbb{E}}[\|\hat{X}_k - \hat{Z}\|^2]$$
$$\leq \frac{2}{\alpha}\,\mathbb{E}_{\hat{\pi}}[\|\nabla^2 V\|_{\mathrm{HS}}^2] + 2\,W_2^2(p_k, \hat{\pi})$$
$$\leq \frac{2d}{\alpha} + 2\,W_2^2(p_k, \hat{\pi}).$$

Collecting the terms,

$$\mathsf{err} \leq \frac{36d}{\alpha} + 6\,W_2^2(p_k, \hat{\pi}).$$

From the assumption $h \leq \frac{\alpha^2}{60}$.

$$\mathbb{E}[W_2^2(p_{k+1}, \hat{\pi}) \mid \mathscr{F}_k] \leq (1 - \alpha h)\,W_2^2(p_k, \hat{\pi}) + \frac{36dh^2}{\alpha}.$$

Iterating this bound proves the result. $\qquad\square$

# F   Proof of Theorem 3

In order to present the proof of Theorem 3, we first review relevant facts about the Wasserstein space over a Riemannian manifold $(\mathcal{M}, \mathfrak{g})$. We refer readers to Villani [2009] for an in-depth treatment.

Similarly to the Euclidean setting, we can define the space of probability measures over $\mathcal{M}$ with finite second moment,

$$\mathcal{P}_2(\mathcal{M}) := \left\{ \mu \in \mathcal{P}(\mathcal{M}) \ \Big| \ \int \mathsf{d}^2(p_0, \cdot)\, \mathrm{d}\mu < \infty \text{ for some } p_0 \in \mathcal{M} \right\},$$

where d denotes the induced distance on $\mathcal{M}$. We equip $\mathcal{P}_2(\mathcal{M})$ with the 2-Wasserstein metric

$$W_2^2(\mu, \nu) := \left[ \inf_{\gamma \in \mathcal{C}(\mu,\nu)} \int \mathsf{d}^2(x, y)\, \mathrm{d}\gamma(x, y) \right]^{1/2},$$

which makes $(\mathcal{P}_2(\mathcal{M}), W_2)$ into a metric space. Moreover, at each regular measure $\mu \in \mathcal{P}_2(\mathcal{M})$, we can define the tangent space

$$T_\mu \mathcal{P}_2(\mathcal{M}) := \overline{\{\nabla \psi \mid \psi \in \mathcal{C}_c^\infty(\mathcal{M})\}}^{L^2(\mu)}$$

equipped with the inner product

$$\langle v, w \rangle_\mu := \int \mathfrak{g}_p\big(v(p), w(p)\big)\, \mathrm{d}\mu(p),$$

which endows $(\mathcal{P}_2(\mathcal{M}), W_2)$ with the structure of a formal Riemannian manifold. Curves $(\mu_t)_{t \geq 0}$ in $\mathcal{P}_2(\mathcal{M})$ are still described the continuity equation

$$\partial_t \mu_t = \mathrm{div}(\mu_t v_t) \tag{46}$$

where now $v_t$ is an element of the tangent bundle $T\mathcal{M}$ and div denotes the divergence operator on the Riemannian manifold. Equation (46) is to be interpreted in the weak sense, i.e., for any test function $\varphi : \mathcal{M} \to \mathbb{R}$,

$$\partial_t \int \varphi\, \mathrm{d}\mu_t = \int \mathfrak{g}(\nabla \varphi, v_t)\, \mathrm{d}\mu_t. \tag{47}$$

If $(\mu_t)_{t \geq 0}$ is a smooth curve such that $\mu_t$ admits a density $\rho_t$ w.r.t. the Riemannian volume measure, then this is equivalent to the partial differential equation (PDE)

$$\partial_t \rho_t = \mathrm{div}(\rho_t v_t).$$

As before, the continuity equation admits a particle interpretation: if $p_0 \sim \mu_0$ and $(p_t)_{t \geq 0}$ evolves via the ODE

$$\dot{p}_t = v_t(p_t), \tag{48}$$

then $p_t \sim \mu_t$ for all $t \geq 0$.

Given a functional $\mathcal{F} : \mathcal{P}_2(\mathcal{M}) \to \mathbb{R} \cup \{\infty\}$ defined over the Wasserstein space, its gradient at $\mu$ is, by definition, the element $\nabla_{W_2} \mathcal{F}(\mu) \in T_\mu \mathcal{P}_2(\mathcal{M})$ such that: for all curves $(\mu_t)_{t \in \mathbb{R}}$ satisfying the continuity equation (46) with $\mu_0 = \mu$, it holds that

$$\partial_t\big|_{t=0} \mathcal{F}(\mu_t) = \langle \nabla_{W_2} \mathcal{F}(\mu), v_0 \rangle_\mu = \int \mathfrak{g}\big(\nabla_{W_2} \mathcal{F}(\mu), v_0\big)\, \mathrm{d}\mu.$$

Using the continuity equation (47), it follows by direct identification that

$$\nabla_{W_2} \mathcal{F}(\mu) = \nabla \delta \mathcal{F}(\mu),$$

where $\delta \mathcal{F}(\mu) : \mathcal{M} \to \mathbb{R}$, the first variation of $\mathcal{F}$ at $\mu$, is defined up to an additive constant and satisfies

$$\partial_t\big|_{t=0} \mathcal{F}(\mu) = \int \delta \mathcal{F}(\mu)\, \partial_t\big|_{t=0} \mu_t.$$

A gradient flow of $\mathcal{F}$ is a curve $(\mu_t)_{t \geq 0}$ which satisfies the continuity equation (46) with velocity vector field $v_t = -\nabla_{W_2} \mathcal{F}(\mu_t)$, which in turn admits the particle interpretation (48).

We now consider the functional

$$\mathcal{F}(\mu) := \mathsf{KL}(\mathsf{p}_\mu \,\|\, \pi)$$

and compute its first variation. Let $\mathfrak{m}$ denote the Riemannian volume measure; let $(\rho_t)_{t\in\mathbb{R}}$ be a smooth curve of densities $\rho_t = \frac{\mathrm{d}\mu_t}{\mathrm{d}\mathfrak{m}}$. Since

$$\mathcal{F}(\mu) = \int V \,\mathrm{d}\mathsf{p}_\mu + \int \mathsf{p}_\mu \ln \mathsf{p}_\mu$$
$$= \iint V \,\mathrm{d}p_\theta \, \rho(\theta) \,\mathrm{d}\mathfrak{m}(\theta) + \iint \ln\Big(\int p_{\theta'} \, \rho(\theta') \,\mathrm{d}\mathfrak{m}(\theta')\Big) \,\mathrm{d}p_\theta \, \rho(\theta) \,\mathrm{d}\mathfrak{m}(\theta)$$

then

$$\partial_t \mathcal{F}(\mu_t) = \iint V \,\mathrm{d}p_\theta \, \dot\rho_t(\theta) \,\mathrm{d}\mathfrak{m}(\theta) + \iint \frac{\int p_{\theta'} \, \dot\rho_t(\theta') \,\mathrm{d}\mathfrak{m}(\theta')}{\int p_{\theta'} \, \rho_t(\theta') \,\mathrm{d}\mathfrak{m}(\theta')} \,\mathrm{d}p_\theta \, \rho_t(\theta) \,\mathrm{d}\mathfrak{m}(\theta)$$
$$+ \iint \ln\Big(\int p_{\theta'} \, \rho(\theta') \,\mathrm{d}\mathfrak{m}(\theta')\Big) \,\mathrm{d}p_\theta \, \dot\rho_t(\theta) \,\mathrm{d}\mathfrak{m}(\theta)$$
$$= \iint (V + \ln \mathsf{p}_{\mu_t} + 1) \,\mathrm{d}p_\theta \, \dot\rho_t(\theta) \,\mathrm{d}\mathfrak{m}(\theta) \,.$$

From this,

$$\delta\mathcal{F}(\mu) : \theta \mapsto \int (V + \ln \mathsf{p}_\mu + 1) \,\mathrm{d}p_\theta = \int \ln \frac{\mathsf{p}_\mu}{\pi} \,\mathrm{d}p_\theta + 1 \,.$$

Next, we compute the Bures–Wasserstein gradient using (33) and (39):

$$\nabla_{\mathsf{BW}}\delta\mathcal{F}(\mu)(m, \Sigma) = \Big(\int \ln \frac{\mathsf{p}_\mu}{\pi} \,\nabla_m p_{m,\Sigma}, \; 2\int \ln \frac{\mathsf{p}_\mu}{\pi} \,\nabla_\Sigma p_{m,\Sigma}\Big)$$
$$= \Big(\int \nabla \ln \frac{\mathsf{p}_\mu}{\pi} \,\mathrm{d}p_{m,\Sigma}, \; \int \nabla^2 \ln \frac{\mathsf{p}_\mu}{\pi} \,\mathrm{d}p_{m,\Sigma}\Big) \,.$$

Finally, to derive the system of ODEs (10), we combine the above expression for the Wasserstein gradient of $\mathcal{F}$ together with the particle interpretation (48) and the equations (31) and (32) for dynamics on the Bures–Wasserstein space.

## G  Lack of convexity of the KL divergence for mixtures of Gaussians

In this section, we provide counterexamples for the lack of convexity of the objective functional $\mu \mapsto \mathcal{F}(\mu) = \mathsf{KL}(\mathsf{p}_\mu \,\|\, \pi)$ on the space $\mathcal{P}_2(\mathsf{BW}(\mathbb{R}^d))$.

First, we point out that even when $\pi$ is strongly log-concave, the functional $\mathcal{F}$ can be badly behaved. For example, if $\pi = p_{0,1} = \mathcal{N}(0, 1)$ is a Gaussian of variance 1, then we can write it as a Gaussian mixture in many ways: $\pi = \int \mathcal{N}(m, a) \,\mathrm{d}\nu_{1-a}(m)$ for any $a \in [0, 1]$, where $\nu_a = \mathcal{N}(0, a)$. In particular, the set of minimizers of $\mathcal{F}$ is not a singleton, and includes all of the measures $\nu_a \otimes \delta_a$ ($(m, \sigma^2)$ is a random pair with independent components, where $m \sim \mathcal{N}(0, 1-a)$ and $\sigma^2 = a$ almost surely) for $a \in [0, 1]$ (as well as all convex combinations—i.e., mixtures—thereof).

Next, we give an explicit example which demonstrates the lack of convexity of the entropy functional $\mu \mapsto \mathcal{H}(\mathsf{p}_\mu) := \int \mathsf{p}_\mu \ln \mathsf{p}_\mu$. This can be understood as the KL divergence with zero potential ($V = 0$). Note that the entropy functional $\mathcal{H}$ is convex on $\mathcal{P}_2(\mathbb{R}^d)$ [Ambrosio et al., 2008, Section 9.4], but our claim is that its composition with the map $\mu \mapsto \mathsf{p}_\mu$ is not convex on $\mathcal{P}_2(\mathsf{BW}(\mathbb{R}^d))$.

In one dimension let $\mu_0 = \mathcal{N}(0, 1) \otimes \delta_1$ and $\mu_1 = \mathcal{N}(0, \tau^2) \otimes \delta_1$. In words, a random pair $(m_0, \sigma_0^2)$ drawn from $\mu_0$ satisfies $m_0 \sim \mathcal{N}(0, 1)$ and $\sigma_0^2 = 1$, and similarly for $\mu_1$. What is the optimal coupling of $\mu_0$ and $\mu_1$? Clearly $\sigma_0^2 = \sigma_1^2 = 1$ is the trivial coupling, and since the Bures–Wasserstein distance over the means is the same as the Euclidean distance between the means, we want the usual $W_2$ optimal coupling between $\mathcal{N}(0, 1)$ and $\mathcal{N}(0, \tau^2)$; it follows that $m_1 = \tau m_0$. Hence, the Bures geodesic between is $\{(m_t, \sigma_t^2) = ((1 - t + t\tau) \, m_0, 1)\}_{t\in[0,1]}$; equivalently the (Bures–)Wasserstein geodesic between $\mu_0$ and $\mu_1$ is $\{\mu_t = \mathcal{N}(0, (1 - t + t\tau)^2) \otimes \delta_1\}_{t\in[0,1]}$.

Next, recall that the Gaussian mixture $\mathsf{p}_{\mu_t}$ is the law of $X$ drawn in the two-stage procedure: first we draw $(m_t, \sigma_t^2) \sim \mu_t$, and given $(m_t, \sigma_t^2)$ we draw $X \sim p_{m_t, \sigma_t^2}$. Thus,

$$\mathsf{p}_{\mu_t} = \int \mathcal{N}(m, \sigma^2) \, \mathrm{d}\mu_t(m, \sigma^2) = \int \mathcal{N}(m, 1) \, \mathrm{d}\nu_{(1-t+t\tau)^2}(m) = \mathcal{N}\big(0, 1 + (1 - t + t\tau)^2\big).$$

Hence,

$$\mathcal{H}(\mathsf{p}_{\mu_t}) = \int \mathsf{p}_{\mu_t} \ln \mathsf{p}_{\mu_t} = -\frac{1}{2} \ln(2\pi e) - \frac{1}{2} \ln\big(1 + (1 - t + t\tau)^2\big).$$

Then, the convexity of $t \mapsto \mathcal{H}(\mathsf{p}_{\mu_t})$ is equivalent to the convexity of $t \mapsto -\ln(1 + (1 - t + t\tau)^2)$, which fails when, e.g., $\tau = 1/2$; in that case, the function is, in fact, concave on the interval $[0, 1]$.

## H  The Wasserstein–Fisher–Rao gradient flow

Similarly to the setting in Section 5, here we identify probability measures $\mu$ over the Bures–Wasserstein space with the corresponding Gaussian mixture $\mathsf{p}_\mu$. The aim of this section is to derive the gradient flow of the KL divergence $\mu \mapsto \mathsf{KL}(\mathsf{p}_\mu \, \| \, \pi)$, except we now equip the space $\mathcal{P}_2(\mathsf{BW}(\mathbb{R}^d))$ with the Wasserstein–Fisher–Rao geometry [Liero et al., 2016, Chizat et al., 2018, Liero et al., 2018]. Deriving the gradient flow with respect to this geometry leads to dynamics for a system of interacting Gaussian particles in which the weight of each particle is also updated at each iteration.

### H.1  Background on Wasserstein–Fisher–Rao geometry

Here we briefly summarize the relevant background on the Wasserstein–Fisher–Rao (WFR) geometry. The WFR metric is also called the *Hellinger–Kantorovich* metric by some authors.

**The Fisher–Rao metric.**  The Fisher–Rao metric is a metric on the space $\mathcal{M}_+(\mathbb{R}^d)$ of positive measures (not necessarily probability measures). It is the induced metric on $\mathcal{M}_+(\mathbb{R}^d)$ if we enforce that the mapping $\mu \mapsto \sqrt{\mu}$ (defined for smooth probability densities $\mu$) is an isometry into $L^2(\mathbb{R}^d)$. This means that

$$\mathsf{d}_{\mathsf{FR}}^2(\mu_0, \mu_1) = \int \big(\sqrt{\mu_0} - \sqrt{\mu_1}\big)^2,$$

and if $\mu_0$ and $\mu_1$ are probability measures then this is known to statisticians (up to a constant factor) as the squared Hellinger distance. (If we apply the analogous procedure to discrete probability measures, then this amounts to identifying the simplex with a subset of the unit sphere.) The Fisher–Rao metric is well-studied in the field of information geometry [Amari and Nagaoka, 2000, Ay et al., 2017].

Next, we describe the Riemannian geometry underlying the Fisher–Rao metric. Consider a curve $t \mapsto \mu_t$ of positive measures with time derivative $\dot{\mu}$. Since the Fisher–Rao metric endows the square root of the density with a Hilbert metric, we place endow the time derivative of the square root, $\dot{\sqrt{\mu}} = \dot{\mu}/(2\sqrt{\mu})$, with the Hilbert norm $\|\dot{\mu}/(2\sqrt{\mu})\|_{L^2(\mathbb{R}^d)}$. Thus, the norm at the tangent space $T_\mu \mathcal{M}_+(\mathbb{R}^d)$ is given by

$$\|\dot{\mu}\|_\mu^2 = \int \frac{\dot{\mu}^2}{4\mu}.$$

Actually, because we are working with positive measures (called *unbalanced* measures to distinguish from the usual optimal transport problem which requires the measures to have the same total mass), this kind of geometry is useful for studying problems in which the total mass changes over time. For example, PDEs of the form $\partial_t \mu_t = \alpha_t \mu_t$ are called reaction equations because they describe, e.g., how the concentration of a chemical changes over time in reaction to the environment. Motivated by this application, we parameterize $\dot{\mu}$ via $\dot{\mu} = \alpha\mu$, in which case the norm is

$$\|\alpha\|_\mu^2 = \frac{1}{4} \int \alpha^2 \, \mathrm{d}\mu. \tag{49}$$

**Wasserstein geometry.** We recall from Section B that Wasserstein geometry is motivated by a completely different class of PDEs, namely *transport equations* encoded by the continuity equation

$$\partial_t \mu_t + \mathrm{div}(\mu_t v_t) = 0 \,,$$

which describe the evolving law of a particle $x_t$ tracing out an integral curve of the family of vector fields: $\dot{x}_t = v_t(x_t)$. The Riemannian structure is obtained by equipping the tangent space $T_\mu \mathcal{P}_2(\mathbb{R}^d)$ with the norm

$$\|v\|_\mu^2 = \int \|v\|^2 \, \mathrm{d}\mu \,.$$

**Wasserstein–Fisher–Rao geometry.** Next we combine the two geometric structures, which can model transport-reaction equations such as

$$\partial_t \mu_t + \mathrm{div}(\mu_t v_t) = \alpha_t \mu_t \,. \tag{50}$$

The tangent space norm is then given by the combination

$$\|(\alpha, v)\|_\mu^2 = \int (\alpha^2 + \|v\|^2) \, \mathrm{d}\mu \,.$$

(At this point some authors add a factor $\frac{1}{4}$ in front of the $\alpha^2$, which is natural in view of (49). This is convenient for studying geometric properties of the space, but it is not necessary for our purposes.) As in the pure Fisher–Rao case, this is a metric on the space of positive measures $\mathcal{M}_+(\mathbb{R}^d)$.

It induces the distance

$$\mathsf{WFR}^2(\mu_0, \mu_1) := \inf\left\{ \int_0^1 \|(\alpha_t, v_t)\|_{\mu_t}^2 \, \mathrm{d}t \ \Big| \ (\mu_t, \alpha_t, v_t)_{t \in [0,1]} \text{ solves (50)} \right\} \,.$$

One can show that the tangent space to $\mathcal{M}_+(\mathbb{R}^d)$ consists of pairs $(\alpha, v)$ for which $\alpha = u$ and $v = \nabla u$ for some function $u : \mathbb{R}^d \to \mathbb{R}$. Thus, compared to the Wasserstein metric in which the tangent space norm is the $\dot{H}^1(\mu)$ norm $\|u\|_{\dot{H}^1(u)} = \|\nabla u\|_{L^2(\mu)}$, the Wasserstein–Fisher–Rao metric has the interpretation of completing the tangent space norm to the full Sobolev norm $H^1(\mu)$.

**Constraining the dynamics to lie within probability measures.** In order to have our dynamics stay on the space of probability measures, we follow Lu et al. [2019] and consider instead the equation

$$\partial_t \mu_t + \mathrm{div}(\mu_t v_t) = \left( \alpha_t - \int \alpha_t \, \mathrm{d}\mu_t \right) \mu_t \,,$$

which now conserves mass. The tangent space norm is modified to read

$$\|(\alpha, v)\|_\mu^2 = \int \left[ \left( \alpha - \int \alpha \, \mathrm{d}\mu \right)^2 + \|v\|^2 \right] \mathrm{d}\mu \,.$$

**Particle interpretation.** The particle interpretation of the WFR geometry is more complicated to state than for the Wasserstein geometry, but it can be done. Instead of considering a particle $x$, we consider a pair $(x, r)$ consisting of a particle $x \in \mathbb{R}^d$ and a number $r > 0$ (this number is actually interpreted as the *square root* of the mass of the particle). The pair $(x, r)$ should be thought of as an element of the cone space $\mathfrak{C}(\mathbb{R}^d) := (\mathbb{R}^d \times \mathbb{R}_+)/(\mathbb{R}^d \times \{0\})$ (in other words, we take the space $\mathbb{R}^d \times \mathbb{R}_+$ and identify all of the points with zero mass which sit at the "tip of the cone"). The cone space is the natural setting for WFR geometry; for example, one can introduce a metric on $\mathfrak{C}(\mathbb{R}^d)$ and show that the WFR distance is an optimal transport problem w.r.t. this metric. We will not go into such detail, but nevertheless we introduce the cone space because is important for the particle interpretation of WFR dynamics.

Curves of measures $(\mu_t)_{t \in [0,1]}$ in the WFR geometry admit a particle interpretation in terms of trajectories on $\mathfrak{C}(\mathbb{R}^d)$. Namely, the equation (50) can be interpreted as follows. There exists a curve of measures $t \mapsto \widetilde{\mu}_t$ over the cone space $\mathfrak{C}(\mathbb{R}^d)$, such that if $r : \mathfrak{C}(\mathbb{R}^d) \to \mathbb{R}_+$ denotes the mapping $(x, r) \mapsto r$, and $x : \mathfrak{C}(\mathbb{R}^d) \to \mathbb{R}^d$ maps $(x, r) \mapsto x$, then

$$\mu_t = x_\#(r^2 \widetilde{\mu}_t) \,.$$

Moreover, if we draw $(x_0, r_0) \sim \widetilde{\mu}_0$ and follow the ODEs

$$\dot{x}_t = v_t(x_t) \,,$$

$$\dot{r}_t = \left( \alpha_t(x_t) - \int \alpha_t \, \mathrm{d}\mu_t \right) r_t \,,$$

then $(x_t, r_t) \sim \widetilde{\mu}_t$. Here the notation $\sim$ is an (egregious) abuse of notation because $\widetilde{\mu}_t$ is not a probability measure; by $(x, r) \sim \widetilde{\mu}$ more precisely we mean that $\widetilde{\mu}_t = (\mathsf{ODE}_t)_\# \widetilde{\mu}_0$ where $\mathsf{ODE}_t$ is the solution mapping $(x_0, r_0) \mapsto (x_t, r_t)$ to the above system of ODEs at time $t$.

To make this interpretation more concrete, we specialize to the case of discrete measures. Suppose that we start at a probability measure

$$\mu_0 = \sum_{i=1}^{N} w_0^{(i)} \delta_{x_0^{(i)}} \,.$$

Then, we lift to the cone space:

$$\widetilde{\mu}_0 = \sum_{i=1}^{N} \delta_{(x_0^{(i)}, \sqrt{w_0^{(i)}})} = \sum_{i=1}^{N} \delta_{(x_0^{(i)}, r_0^{(i)})}$$

where we set $r_t^{(i)} = \sqrt{w_t^{(i)}}$. Next, we follow the ODEs

$$\dot{x}_t^{(i)} = v_t(x_t^{(i)}) \,,$$

$$\dot{r}_t^{(i)} = \left( \alpha_t(x_t^{(i)}) - \sum_{j=1}^{N} w_t^{(j)} \alpha_t(x_t^{(j)}) \right) r_t^{(i)} \,.$$

Upon projecting back to the base space, we obtain another discrete measure

$$\mu_t = \sum_{i=1}^{N} w_t^{(i)} \delta_{x_t^{(i)}} = \sum_{i=1}^{N} (r_t^{(i)})^2 \, \delta_{x_t^{(i)}} \,.$$

As a sanity check, we check that these dynamics ensure that $\mu_t$ is a probability measure for all $t$. The time derivative of the sum of the weights is

$$\partial_t \sum_{i=1}^{N} w_t^{(i)} = 2 \sum_{i=1}^{N} r_t^{(i)} \, \partial_t r_t^{(i)} = 2 \sum_{i=1}^{N} (r_t^{(i)})^2 \left( \alpha_t(x_i^{(t)}) - \mathbb{E}_{\mu_t} \alpha_t \right)$$

$$= 2 \left( \sum_{i=1}^{N} w_t^{(i)} \alpha_t(x_i^{(t)}) - \mathbb{E}_{\mu_t} \alpha_t \right) = 0 \,.$$

## H.2  Derivation of the gradient flow

Next, we derive the Wasserstein–Fisher–Rao gradient flow of the functional $\mu \mapsto \mathcal{F}(\mu) := \mathsf{KL}(\mathsf{p}_\mu \,\|\, \pi)$ on the space $(\mathcal{P}_2(\mathsf{BW}(\mathbb{R}^d)), \mathsf{WFR})$ of Gaussian mixtures equipped with the Wasserstein–Fisher–Rao metric (over the Bures–Wasserstein space). The WFR gradient of $\mathcal{F}$, $\nabla_{\mathsf{WFR}} \mathcal{F}(\mu)$, is the pair

$$\nabla_{\mathsf{WFR}} \mathcal{F}(\mu) = \left( \nabla_{\mathsf{BW}} \delta \mathcal{F}(\mu), \; \delta \mathcal{F}(\mu) - \int \delta \mathcal{F}(\mu) \, \mathrm{d}\mu \right) .$$

This result is essentially stated as Lu et al. [2019, Proposition A.1], although we have generalized the formula to hold when the base space is no longer $\mathbb{R}^d$. Note also that we have already calculated the first variation of $\mathcal{F}$, as well as the BW gradient, in Section F.

The interpretation of the formula is that in the gradient flow of $\mathcal{F}$, we have a particle $(m, \Sigma)$ associated with some mass $w$ evolving according to

$$\dot{m} = - \mathbb{E}_{p_{m,\Sigma}} \nabla \ln \frac{\mathsf{p}_\mu}{\pi} \,,$$

$$\dot{\Sigma} = - \Sigma \, \mathbb{E}_{p_{m,\Sigma}} \nabla^2 \ln \frac{\mathsf{p}_\mu}{\pi} - \mathbb{E}_{p_{m,\Sigma}} \nabla^2 \ln \frac{\mathsf{p}_\mu}{\pi} \, \Sigma \,,$$

$$\dot{r} = - \left( \mathbb{E}_{p_{m,\Sigma}} \ln \frac{\mathsf{p}_\mu}{\pi} - \mathbb{E}_{\mathsf{p}_\mu} \ln \frac{\mathsf{p}_\mu}{\pi} \right) r \,,$$

where $r = \sqrt{w}$. The interpretation may be clearer in the discrete case, so suppose that we initialize the dynamics at a discrete measure

$$\mu_0 = \sum_{i=1}^{N} w_0^{(i)} \delta_{(m_0^{(i)}, \Sigma_0^{(i)})} \,.$$

Next we solve the coupled system of ODEs, for $i \in [N]$,

$$\dot{m}^{(i)} = -\mathbb{E}_{p_{m^{(i)}, \Sigma^{(i)}}} \nabla \ln \frac{\mathsf{p}_\mu}{\pi} \,,$$

$$\dot{\Sigma}^{(i)} = -\Sigma^{(i)} \mathbb{E}_{p_{m^{(i)}, \Sigma^{(i)}}} \nabla^2 \ln \frac{\mathsf{p}_\mu}{\pi} - \mathbb{E}_{p_{m^{(i)}, \Sigma^{(i)}}} \nabla^2 \ln \frac{\mathsf{p}_\mu}{\pi} \Sigma^{(i)} \,,$$

$$\dot{r}^{(i)} = -\left( \mathbb{E}_{p_{m^{(i)}, \Sigma^{(i)}}} \ln \frac{\mathsf{p}_\mu}{\pi} - \mathbb{E}_{\mathsf{p}_\mu} \ln \frac{\mathsf{p}_\mu}{\pi} \right) r^{(i)} \,,$$

where $r^{(i)} = \sqrt{w^{(i)}}$ and

$$\mu_t = \sum_{i=1}^{N} w_t^{(i)} \delta_{(m_t^{(i)}, \Sigma_t^{(i)})} \,.$$

Since the normalization constant of $\pi$ cancels out in the above equations, they are implementable without this knowledge.

# I  Experiments for Gaussian VI

The goal of the present section is to conduct numerical experiments that illustrate the convergence of the Gaussian distribution corresponding to the ODE (4) to an approximation of the target distribution. We consider two kinds of targets: a mixture of two Gaussians, and a log-concave target that corresponds to the likelihood function in logistic regression.

## I.1  Setup

### I.1.1  Definition of the target distributions

**Bimodal target: mixture of two Gaussians**
We define a bimodal target as a mixture of two Gaussians $\pi = \frac{1}{2} \mathcal{N}(\mu_1, \Sigma_1) + \frac{1}{2} \mathcal{N}(\mu_2, \Sigma_2)$ where $\Sigma_1$ and $\Sigma_2$ have non isotropic covariances with a ratio of 3 between the largest and the smallest eigenvalues.

**Log-concave target: Bayesian logistic regression**
The proposed log-concave target is generated in the context of the Bayesian treatment of logistic regression associated with a two-class synthetic dataset $\mathcal{D} = \{(x_i, y_i) : i = 1, \dots, N\}$. The probability of the binary label $y_i \in \{0, 1\}$ given the corresponding covariate $x_i$ and parameter $z \in \mathbb{R}^d$ is defined by the following Bernoulli distribution:

$$\pi(y_i | x_i, z) = \sigma(x_i^\mathsf{T} z)^{y_i} (1 - \sigma(x_i^\mathsf{T} z))^{1 - y_i} \,, \tag{51}$$

where $\sigma(x) = 1/(1 + \exp(-x))$ is the logistic function. We define the target distribution as the posterior associated to data $\mathcal{D}$ starting from an uninformative (flat) prior on $z$, that is,

$$\pi(z | \mathcal{D}) = \frac{1}{Z} \prod_{i=1}^{N} \pi(y_i | x_i, z) \,, \tag{52}$$

with $Z$ the normalization constant. The Langevin dynamics are associated with the gradient of $V$ then defined by:

$$-\nabla V(z) = \nabla \log \pi(z | \mathcal{D}) = \sum_{i=1}^{N} (y_i - \sigma(x_i^\mathsf{T} z)) \, x_i \,.$$

To generate the synthetic data $\mathcal{D}$, we randomly draw labels $y_i \in \{0, 1\}$ and for the problem to be well-specified we have drawn the class-conditional covariates $x_i$ from Gaussian distributions $\mathcal{N}(m_{y_i}^*, \Sigma^*)$ with $m_1^* = -m_0^* = m^*$. We call $s$ the separation factor defined by $\|m_1^* - m_0^*\| = \|2m^*\| = s$. For illustrative purposes we also plot Fisher's linear discriminant vector defined by $z^* = 2\Sigma^{*-1} m^*$ [see Bishop, 2006, chapter 4]. An example of the generated data is displayed in Figure 4.

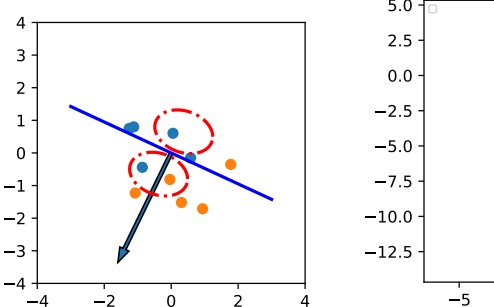

Figure 4: The dataset $\mathcal{D}$ (left figure) used to generate the target distribution (right figure). The two Gaussians of equal covariance $\Sigma^*$ from which the covariates are generated are shown as red ellipsoids. The arrow represents Fisher's linear discriminant $z^* = 2\Sigma^{*-1}m^*$.

### I.1.2  Evaluation of the KL divergence for the proposed log-concave targets

The target distribution (52) may be written $\pi(z|\mathcal{D}) = \frac{1}{Z}\,\tilde{\pi}(z|\mathcal{D})$ where $\tilde{\pi}(z|\mathcal{D}) = \prod_{i=1}^{N}\pi(y_i|x_i,z)$ is the unnormalized distribution. The divergence between any Gaussian distribution $p = \mathcal{N}(m,\Sigma)$ and the target then writes

$$\mathsf{KL}(p(z)\,\|\,\pi(z|\mathcal{D})) = \int p(z)\ln\frac{p(z)}{\pi(z|\mathcal{D})}\,\mathrm{d}z = \int p(z)\ln\frac{p(z)}{\tilde{\pi}(z|\mathcal{D})}\,\mathrm{d}z + \ln Z \qquad (53)$$

$$= -\int p(z)\ln\tilde{\pi}(z|\mathcal{D})\,\mathrm{d}z + H(p) + \ln Z \qquad (54)$$

where $H(p)$ is the negative entropy of a Gaussian distribution for which a closed-form expression is known. We will see shortly we can approximate the expectation under the Gaussian $p$ as follows

$$\mathsf{KL}(p(z)\,\|\,\pi(z|\mathcal{D})) \approx -\sum_{i=1}^{K}\alpha_i\ln\tilde{\pi}(m + c_i R e_i|\mathcal{D}) + H(p) + \ln Z\,, \qquad (55)$$

using $K = 2d$ sigma points with cubature rules $(\alpha_i, c_i) = (\frac{1}{2d}, \sqrt{d})$ for all $i$, and where $R$ is defined via the Cholesky decomposition $RR^{\mathsf{T}} = \Sigma$ (see Section I.2 for details).

### I.1.3  The Laplace approximation as a baseline

We use the widespread Laplace approximation [see Bishop, 2006, chapter 4] as a baseline for comparisons. In dimension 2, we compute the normalization constant $Z$ of (52) using a grid. When we turn to high dimension, normalization becomes intractable. However, we may still compare our algorithm with Laplace approximation as follows. Since our goal is mainly to illustrate the convergence of our algorithm using Laplace approximation as a baseline, we may choose an arbitrary value for the normalization constant $Z$ when evaluating the divergence to the target in equation (55). This allows for comparison of the KL divergence between the approximating distribution—given by either Gaussian VI or Laplace approximation—and the target $\pi$ up to the same additive constant for both methods. By default, we let $Z = 1$, but we sometimes use larger values of $Z$ in order to avoid plotting negative values for the unnormalized KL (albeit an arbitrary choice).

To obtain the Laplace approximation, we first compute a mode of the target distribution $\pi$. Once the mode $z_0$ has been found, we consider the following Taylor approximation around the mode:

$$\ln\pi(z) \approx \ln\pi(z_0) - \frac{1}{2}(z - z_0)^{\mathsf{T}}H(z - z_0)\,, \qquad (56)$$

where $H$ is the Hessian of the negative log-likelihood around $z_0$ defined by $H = \nabla^2\log\frac{1}{\pi}(z_0)$. Renormalizing, this yields the approximation

$$\pi \approx \hat{\pi}^{\mathsf{Laplace}} = \mathcal{N}(z_0, H^{-1})\,. \qquad (57)$$

In our experiments, we use the L-BFGS algorithm [Liu and Nocedal, 1989] to find the mode $z_0$.

## I.2 Implementation

We follow Särkkä [2007], Lambert et al. [2022b] to compute the expectations involved in equation (4) using quadrature rules. We then numerically integrate the set of coupled ODEs in equation (4) using a fourth-order Runge–Kutta method. As a first step, we introduce a method to enforce that the covariance matrix $\Sigma$ remains symmetric and positive at all times.

- **Covariance matrices in square root form:** To numerically enforce that the covariance matrix $\Sigma$ remains symmetric and positive at each step, as is customary in the Kalman filtering literature, we consider a continuous-time "square-root" form of the covariance as developed in Morf et al. [1977] and applied in Särkkä [2007]. Let $R$ be a lower triangular matrix such that $\Sigma = RR^\mathsf{T}$. An ODE for $R$ is obtained as follows.

$$\dot{\Sigma} = \dot{R}R^\mathsf{T} + R\dot{R}^\mathsf{T} \tag{58}$$

  Multiplying by $R^{-1}$ on the left and $R^{-\mathsf{T}}$ on the right yields:

$$R^{-1}\dot{R} + \dot{R}^\mathsf{T}R^{-\mathsf{T}} = R^{-1}\dot{\Sigma}R^{-\mathsf{T}}. \tag{59}$$

  As $R^{-1}\dot{R} + \dot{R}^\mathsf{T}R^{-\mathsf{T}} = R^{-1}\dot{R} + (R^{-1}\dot{R})^\mathsf{T}$, the solution is given by:

$$R^{-1}\dot{R} = \mathrm{Tria}(R^{-1}\dot{\Sigma}R^{-\mathsf{T}}), \tag{60}$$

$$\dot{R} = R\,\mathrm{Tria}(R^{-1}\dot{\Sigma}R^{-\mathsf{T}}), \tag{61}$$

  where $\mathrm{Tria}(A)$ gives the lower triangular matrix $L$ corresponding to $A$ such that $A = L+L^\mathsf{T}$ where $L_{i,i} = \frac{1}{2}A_{i,i}$, $L_{i,j} = A_{i,j}$ if $i > j$, and $L_{i,j} = 0$ otherwise. Letting $\dot{\Sigma}$ be as in (4), this yields an ODE in terms of the square root factor $R$.

- **Computing expectations:** We compute Gaussian expectations using a quadrature rule based on $2d$ sigma points $x_1, \ldots, x_{2d}$ [Julier and Uhlmann, 2004]:

$$\mathbb{E}_{p_{m,\Sigma}}[f(x)] \approx \sum_{n=1}^{2d} \alpha_n f(x_n),$$

  where the sigma points are distributed according to $x_n = m + c_n R e_n$, where $RR^\mathsf{T} = \Sigma$, $e_n|_{n=1,\ldots,d}$ is a basis, and $e_n|_{n=d+1,\ldots,2d}$ is its negative. Many variants exist to choose $\alpha_n$ and $c_n$; here, we consider the cubature points of Arasaratnam and Haykin [2009] defined by $\alpha_n = \frac{1}{2d}$ and $c_n = \sqrt{d}$ which are well-adapted for Gaussian integration.

## I.3 Results in dimension 2

We first conduct experiments in dimension 2 to easily visualize the true posterior (normalization is performed using a discrete grid of size $100 \times 100$).

### I.3.1 Trajectories generated by numerical integration of the ODEs

In Figure 5, we see that Gaussian VI converges quickly to one mode of the bimodal target, and to the unique mode of the logistic target. As shown in Figure 6, the results still hold if we choose a larger step size for the Runge–Kutta scheme.

### I.3.2 Comparison with the Laplace approximation

We compare Gaussian VI with the Laplace approximation on the logistic target in dimension 2 for the setting described in Section I.1 with an arbitrary $\Sigma^*$ and $N = 10$. We plot the convergence speed of our algorithm for Gaussian VI in Figure 7 for separation parameters $s = 1.5$ and $s = 2$, the latter corresponding to a sharper density. Gaussian VI converges very fast and produces a better approximation of the target in terms of KL divergence than the Laplace approximation.

## I.4 Results in higher dimensions

We now compare Gaussian VI with the Laplace approximation on the logistic target in dimension $d = 10$ and $d = 100$. We consider the setting described in Section I.1 where we let $\Sigma^* = \frac{1}{d}I$, to have consistent norms of the inputs accross dimensions.

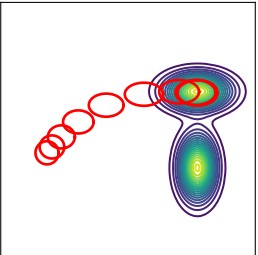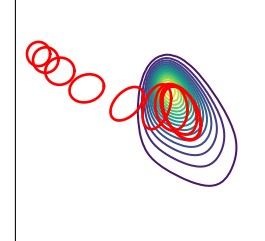

Figure 5: Approximation of a bimodal target (left) and a logistic target (right). We use a Runge–Kutta scheme with step size $0.1$ and a time duration of $T = 30$ (i.e., 300 steps). The ellipsoids represent the Gaussian computed at successive steps.

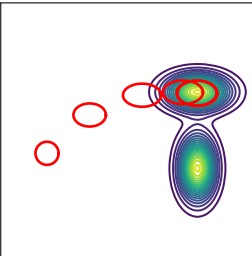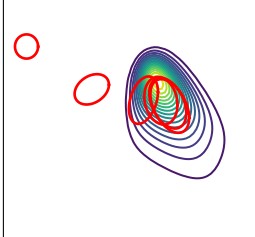

Figure 6: Same as Figure 5 with a larger Runge–Kutta step size 1 (i.e., 30 steps). In both cases, the algorithm converges to the same approximation as in Figure 5.

For Gaussian VI in high dimension, we find that a step size 1 for the Runge–Kutta integration method is too large and leads to singular covariance matrices. We thus take the step size equal to $0.1$. The initial Gaussian is taken to be $\mathcal{N}(0, 100I)$, to better cover regions of low density initially.

Results are shown in Figures 8 and 9 in dimension $d = 10$ and 100 respectively. Gaussian VI converges very fast and always produces a better approximation of the target in terms of KL divergence than the Laplace approximation. Note that the Laplace approximation can have a very high left KL divergence when the target distribution is sharp (i.e., when the two classes are well-separated). This is because the Gaussian approximation computed with the Laplace method tends to spill out of the target distribution in region of very low densities.

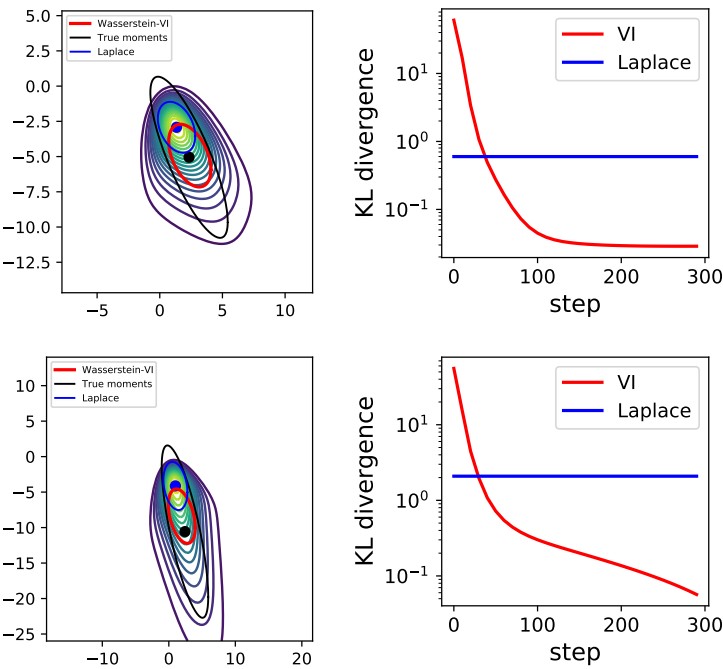

Figure 7: Results in dimension $d = 2$, $N = 10$ for a separation factor $s = 1.5$ (upper row) and $s = 2$ (lower row). The left column shows the true density via contour lines, the true mean (black dot) and covariance (black ellipsoid), and the results of the Laplace and Wasserstein VI approximations as blue and red ellipsoids respectively. The right column shows the evolution of the left KL divergence for Gaussian VI on a logarithmic scale. The corresponding KL divergence obtained with Laplace approximation is shown as a blue straight line.

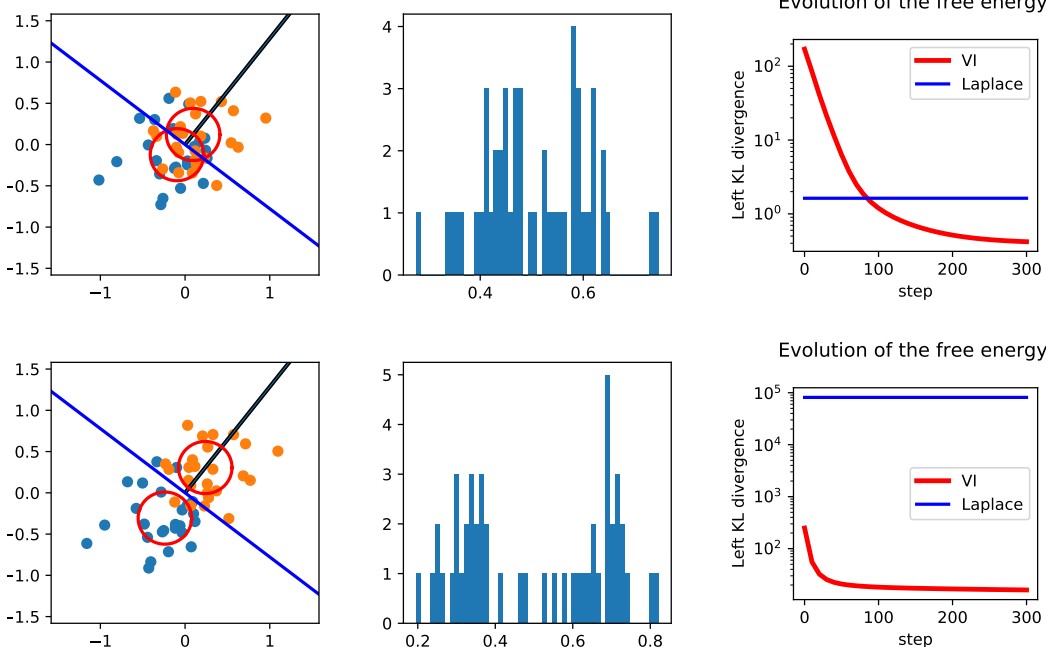

Figure 8: Results in dimension $d = 10$, $N = 50$ for a separation factor $s = 0.6$ (upper row) and $s = 1.5$ (lower row). Left column: synthetic dataset projected onto the two first coordinates. Middle column: histogram representing the number of examples predicted at a given probability by the obtained classifier. Right column: convergence in terms of unnormalized KL divergence. The unnormalized KL is computed via (55) letting $Z = 1$ (upper row) and $Z = 10^{20}$ (lower row). The Runge–Kutta step size is set to $0.1$.

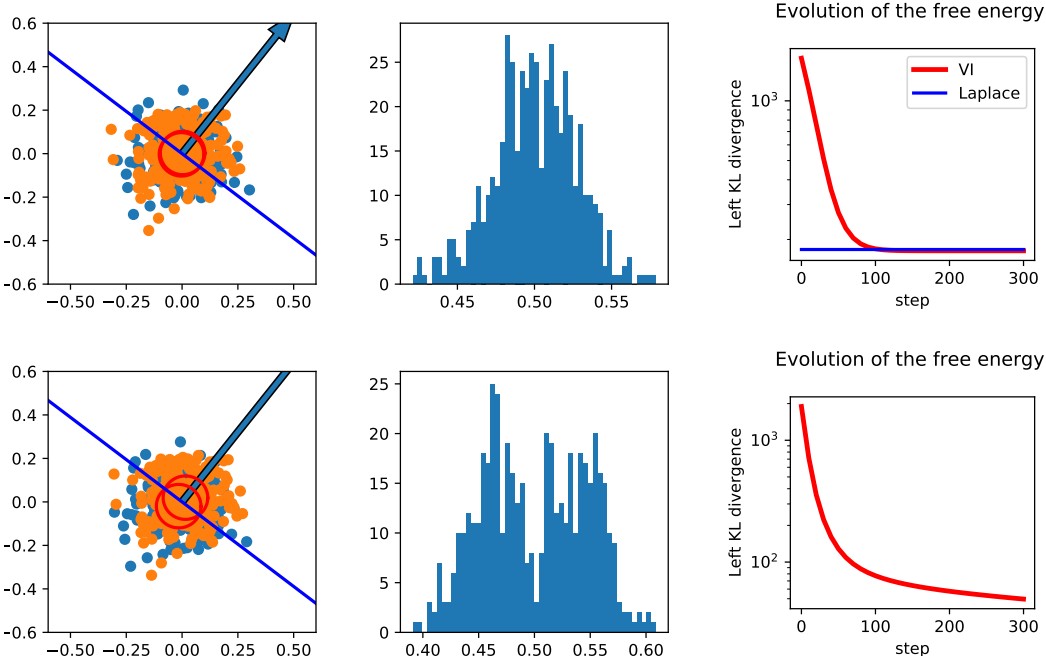

Figure 9: Same as Figure 8 but with dimension $d = 100$, $N = 500$, with separation factor $s = 0.05$ (upper row) and $s = 0.3$ (lower row). The unnormalized KL is computed letting $Z = 1$ (upper row) and $Z = 10^{100}$ (lower row). The unnormalized KL divergence for the Laplace method is not shown in the lower plot because it is too large to be visualized.

# J Experiments for mixture of Gaussians VI

In this section, we consider a mixture of Gaussians model to approximate a target distribution in the simple two-dimensional case. The goal is to illustrate the convergence of the approximating particles system (11)-(12) to an approximation of the target in the form of a finite mixture of Gaussians.

## J.1 Setup

We consider the bimodal and logistic targets defined in Section I, as well as more complex targets defined as finite mixtures of Gaussians:

$$\pi = \sum_{i=1}^{M} w_i^* \, \mathcal{N}(m_i^*, \Sigma_i^*) \,.$$

The gradient $\nabla_x \log \pi(x)$ then writes:

$$\nabla_x \log \pi(x) = \frac{1}{\pi(x)} \, \nabla_x \pi(x) = \frac{1}{\pi(x)} \sum_{i=1}^{M} w_i^* \, \Sigma_i^{*-1} \, (x - m_i^*) \, \mathcal{N}(x \mid m_i^*, \Sigma_i^*) \,.$$

We consider $K$ Gaussian samples equally weighted such that our mixture model is $p = \frac{1}{K} \sum_{i=1}^{K} p_i = \frac{1}{K} \sum_{i=1}^{K} \mathcal{N}(m_i, \Sigma_i)$. Even if we are using an approximation with equal weights, contrary to the target (which can be arbitrary in practice), we can hope from Theorem 3 convergence to a good approximation of $\pi$ when letting $K \gg M$.

## J.2 Implementation details

### J.2.1 Integration of the ODEs

Following equations (11)-(12), we implement the system of ODEs

$$\dot{m}_k = \mathbb{E}_{p_k}[\nabla_x \ln \pi] - \mathbb{E}_{p_k}[\nabla_x \ln p] \,,$$
$$\dot{\Sigma}_k = A + A^\mathsf{T} \,,$$
$$\text{where } A = \mathbb{E}_{p_k}[(x - \mu_k) \otimes \nabla_x \ln \pi] - \mathbb{E}_{p_k}[(x - \mu_k) \otimes \nabla_x \ln p] \,.$$

We recall that these equations arise from applying Theorem 3 to a discrete mixing measure and applying integration by parts to obtain Hessian-free updates. To constrain the covariance matrix to remain definite positive along the numerical integration process, we use the same method as in the Gaussian VI case (Section I.2): we replace each ODE for a covariance matrix $\Sigma$ by an ODE for its lower triangular matrix factor $R$ where $\Sigma = RR^\mathsf{T}$. To compute the expectations, we use the sigma points with cubature rules as described in Section I.2.

Finally, to solve the ODEs we consider a classical Runge–Kutta scheme of $4^{\text{th}}$ order. The coupling between the ODEs is taken into account by applying the Runge–Kutta algorithm on the joint ODE $\dot{X} = F(X)$ where the Gaussian parameters are stacked as follows: $X = [m_1, \ldots, m_K, \text{vec}(R_1), \ldots, \text{vec}(R_K)]$. For our problem, setting the Runge–Kutta step size to $0.1$ is sufficient. We observe that asymptotic convergence, i.e., complete stability of the ODE system, may require many iterations when we propagate a large number of coupled Gaussian particles. On the other hand, the KL divergence is roughly stable after 30 steps.

### J.2.2 Initialization of the Gaussian particles

We start by illustrating the sensitivity of the algorithm to the initialization on a simple example with one Gaussian particle and a bimodal target (Figure 10). When the initial particle is close to one of the two modes and has same covariance as each mode, then it moves towards that mode and its covariance remains constant. When the particle is equidistant from the two modes, then the mean of the particle converges to the average of the two modes, and its covariance increases. Perturbing the initial condition slightly leads the particle to be attracted to one of the two modes.

To avoid bad initialization, the idea is to generate instead more particles than the number of modes of the target. Finally, we initialize our Gaussian particles with means randomly chosen from a Euclidean ball which covers most of the mass of the target density.

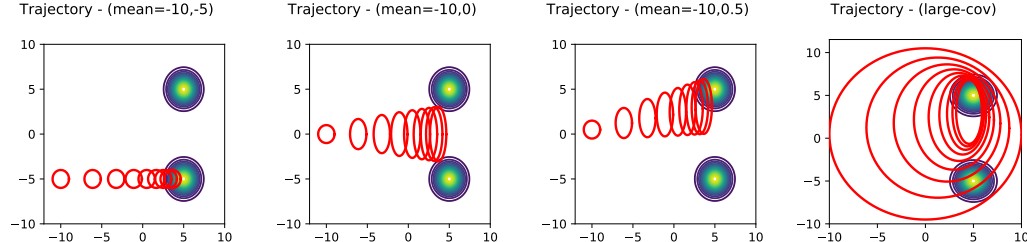

Figure 10: Trajectory of a Gaussian particle for different initial conditions. In the three left plots, we initialize the particle with the same covariance as each mode, and in the right plot we initialize the particle with a large covariance.

## J.3 Experimental results

We show qualitative fits by plotting the contour lines of the approximated density (compared to the true density), as well as quantitative evaluation of the KL divergence to the target.

The true posterior is computed using a discrete grid of size $100 \times 100$. The KL divergences are evaluated using Monte Carlo sampling.

### J.3.1 Simple targets

We consider a mixture of 20 Gaussians to approximate the targets defined in Section I.1. We see in Figure 11 that the algorithm captures both modes of the bimodal distribution, and approximates well the logistic target also, see Figure 12.

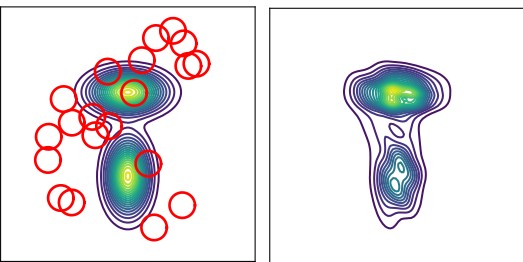

Figure 11: Approximation of the bimodal target using 20 Gaussian particles at initialization (left) and at final step (right). We use Runge–Kutta integration with step size 0.1 and integrattion time $T = 30$ (i.e., 300 steps).

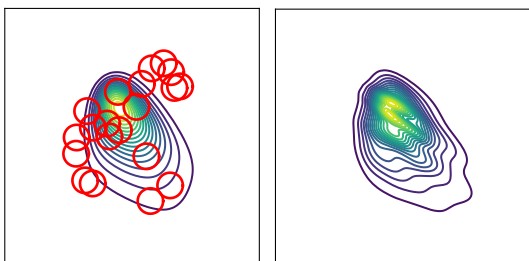

Figure 12: Approximation of the logistic target with 20 Gaussian particles. We use Runge–Kutta integration with step size 0.1 and integration time $T = 30$ (i.e., 300 steps).

### J.3.2 More complex targets

We assess the sensitivity to the number of particles in Figures 13, 14, and 15. When the number of particles increases, better KL divergence is achieved and the distribution is better approximated.

We also note that when the samples initially cover a low density mode as in Figure 14, they tend to overestimate the local density before they escape the mode.

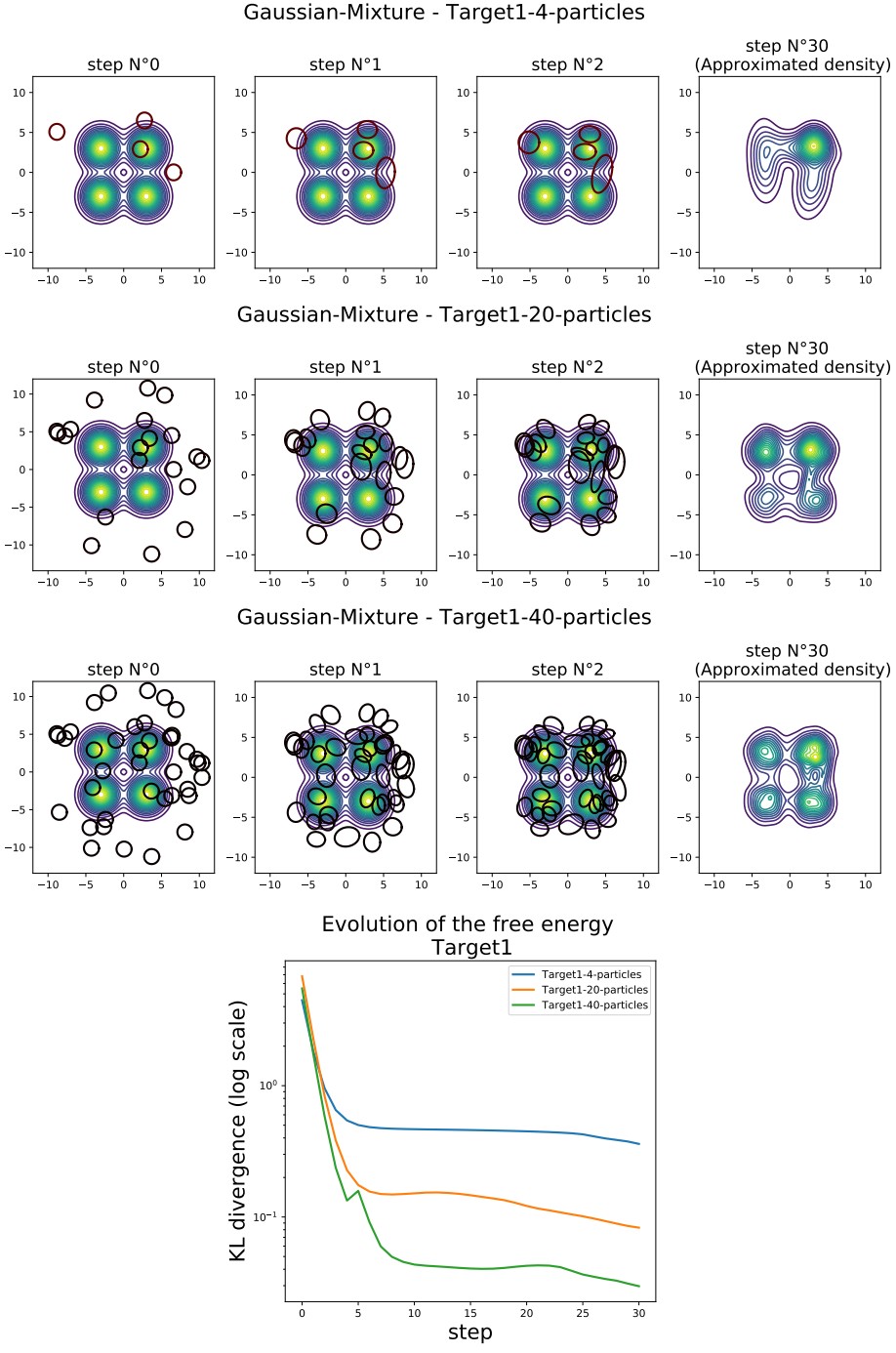

Figure 13: A target with 4 equally weighted modes and isotropic covariances.

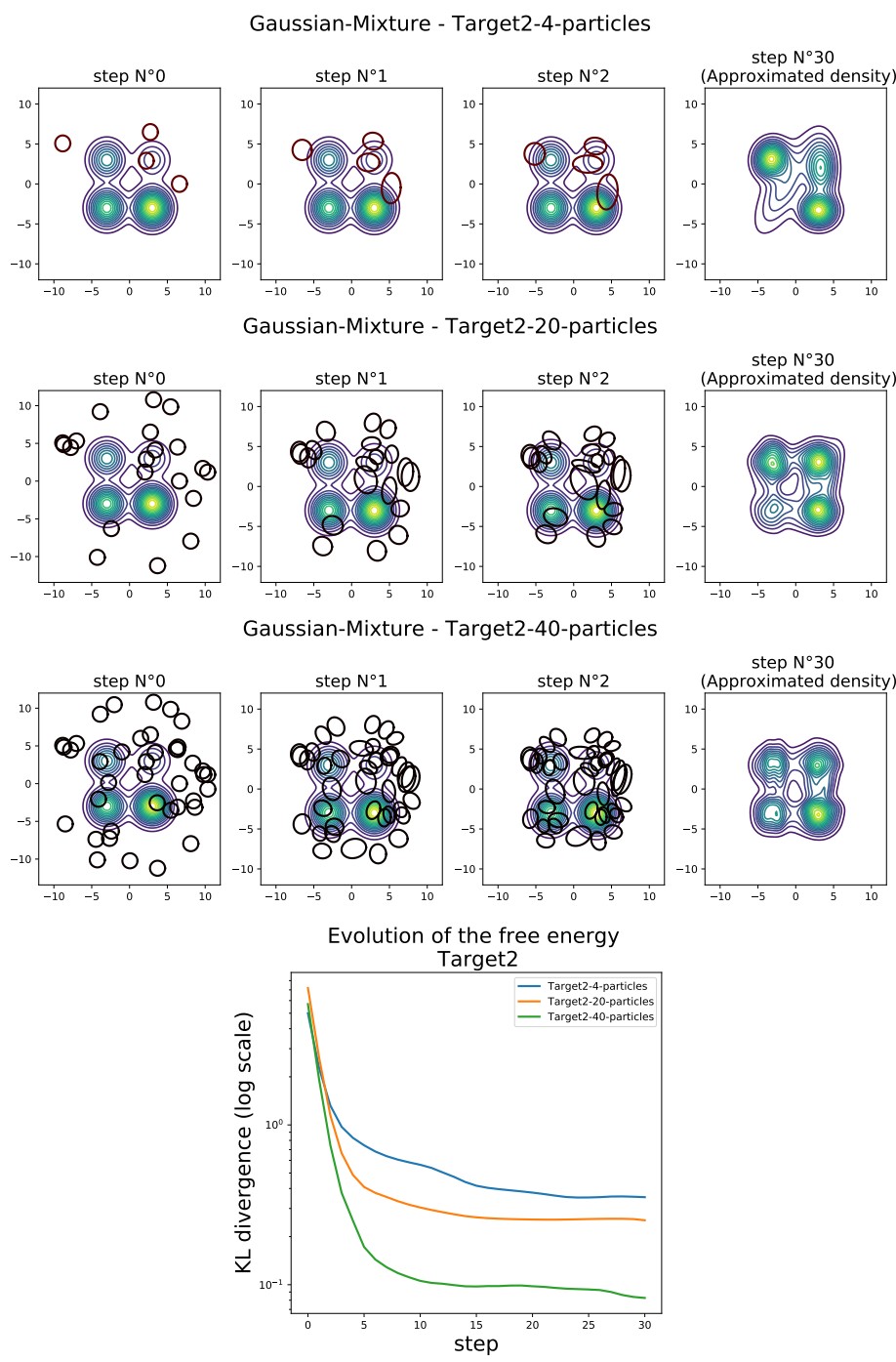

Figure 14: A target with 4 non-equally weighted modes and isotropic covariances.

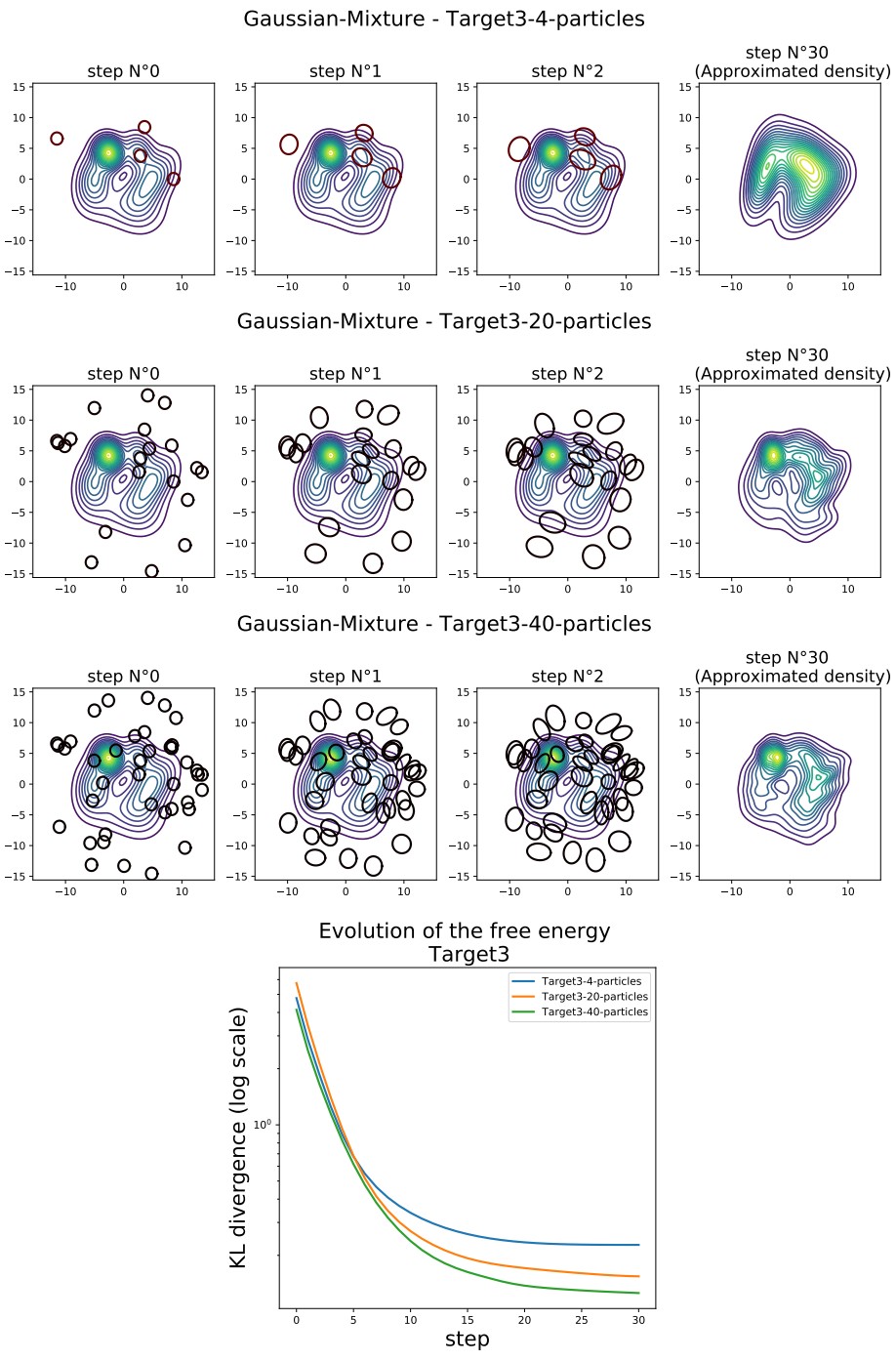

Figure 15: A target with 6 non-equally weighted modes and isotropic covariances.