# OpenReview forum: "Variational inference via Wasserstein gradient flows"
_NeurIPS.cc/2022/Conference — NeurIPS 2022 Accept_

### Official Review · Reviewer_E7dv · 2022-07-07

**Rating:** 8
**Confidence:** 4
**Soundness:** 4 excellent
**Presentation:** 4 excellent
**Contribution:** 4 excellent

**Summary:**

This paper produces an algorithm for computing variational posteriors that are normals or mixtures of normals. As the title suggests, this is achieved via Wasserstein gradient flows. Remarkably, the system of the Langevin-type stochastic process this gives rise to was proposed before---but as a heuristic in the setting of sequential bayesian inference/kalman filtering. Relative to existing variational methods, the benefits of the proposed approach are a number of theoretical guarantees that are usually not available, and hold under reasonable (even if somewhat limiting) assumptions so long as one uses a single normal (rather than a mixture of normals) as the variational family.

**Questions:**

I would advise the authors to tone down the claims about VI being a mostly heuristic field without significant theoretical advances; and to embedd their paper into the context of [1-4] and related work provided above.

**Limitations:**

I think that the paper does not point to the limitations of its results enough, but I don't think this is such a big problem---there is only so much space a NeurIPS submission is allowed to cover, and the paper does a better job than most others. The three most important limitations are
(1) V has to be convex
(2) guarantees only hold for the single Gaussian measure case; and not even for the mixture Gaussian case [and to be fair, this one is pointed out explicitly!]
(3) it is unclear if the algorithm is actually faster/better than standard VI techniques/optimisation methods in practice

**Strengths And Weaknesses:**

Originality:
The work is highly original from a technical point of view, & to the best of my knowledge the first attempt at using Wasserstein Gradient flows in a computationally attractive way for the purposes of variational inference. Its originality lies not so much in the end goal of the algorithm---there are plenty other variational methods using Gaussian families/mixtures of Gaussians---but in using an area of mathematics that has hitherto seen little use in the context of variational inference to derive a new algorithm for achieving an (old/well-known) end goal and guaranteeing a number of attractive properties for the resulting algorithm.

Quality:
This paper is of the highest technical standard. Beyond that, the authors have made a very serious effort to show that the method is not just a mathematical curiosity, but practically meaningful. It has been a while since a paper has inspired me to dive into a different literature and read around 5-10 other papers to fully appreciate all its details, but this paper has managed it effortlessly.

Clarity:
The paper articulates its (complicated) subject clearly, and leaves the reader satisfied that she or he can understand every component part of the paper. The supplementary material is extensive, and maintains an exceptionally high standard.

Significance:
I think the paper is significant in showing how to use Wasserstein gradient flows for the design of attractive variational methods. If I have one point of criticism towards the paper, it would be that the research within the field of variational inference is misrepresented. For example, the paper's abstract states that "VI is still poorly understood and dominated by heuristics. In this work, we propose principled methods for VI [...]". I think this sentiment does severe injustice to a lot of good work that has happened on VI in recent times. I understand that the authors are focused specifically on optimisation algorithms with theoretical guarantees that are common in optimisation, but other theoretical work from other fields and specifically statistics has recently put VI on a much more solid foundation than it used to be. To name but a few examples, there are PAC-Bayesian bounds for variational objectives [1] as well as asymptotic studies [2], work motivating variational methods as constrained optimisation [3], and even some prior work on convergence guarantees (under conditions that are in some sense milder than those of the current paper) [4]. I think the paper would do good toning down the claims of the abstract & the other claims in the paper aligned with the spirit of the abstract---or at least make them more specific to the kind of guarantees/theory that the paper is focused on. It certainly isn't accurate to describe all of the research on VI as 'dominated by heuristics'. I think the paper would also do well positioning itself in the context of a recent line of work reaching across various disciplines and dedicated to making VI a more principled set of tools as in [1-4].

[1] https://jmlr.org/papers/volume17/15-290/15-290.pdf

[2] https://www.tandfonline.com/doi/full/10.1080/01621459.2018.1473776

[3] https://www.jmlr.org/papers/v23/19-1047.html

[4] https://proceedings.mlr.press/v119/domke20a.html


Typos:
- p.7, l.270 'variation' -> 'variational'

---

> ### Author Response · Authors · 2022-08-01
> **Response**
>
> Thank you for your kind review. We are glad that you enjoyed reading our submission.
>
> > If I have one point of criticism towards the paper, it would be that the research within the field of variational inference is misrepresented.
>
> Thank you for bringing up this omission. As we note in our response to all reviewers, we only became aware of this literature after submission, and many of the inappropriate sentences of the paper regarding VI stemmed from this omission. We will conduct a more thorough literature search, provide a discussion of the placement of our paper in the context of this work, and soften some of our claims accordingly.

---

> > ### Comment · Reviewer_E7dv · 2022-08-04
> > **Thank you; I appreciate your commitment to improving your excellent work further.**
> >
> > Thank you; I appreciate your commitment to improving your excellent work further.

---

### Official Review · Reviewer_8QPd · 2022-07-08

**Rating:** 6
**Confidence:** 4
**Soundness:** 3 good
**Presentation:** 3 good
**Contribution:** 2 fair

**Summary:**

This paper proposes a variational inference method to find the optimal approximation of the target distribution, which has the smallest KL divergence to the target distribution in the family of Gaussian distribution or Gaussian mixture. For the family of Gaussian distribution, the convergence analysis of the continuous-time dynamics and the discrete-time stochastic gradient algorithms are established under the strong log-concavity assumption of the target density. For the Gaussian mixture, they propose the gradient flow of the KL divergence and the particle discretization by parametrizing the Gaussian mixture model as a probability measure over the Bures-Wasserstein space.


**Questions:**

The interpretation of the Gaussian mixture model as the measure on the Bures-Wasserstein space is interesting. I’m wondering if the target density itself is a Gaussian mixture, can the proposed algorithm exactly fit the posterior distribution?

For the Gaussian mixture, are there objective functionals other than the KL divergence, for instance, maximum mean discrepancy, to make the resulting problem on the measure of the Bures-Wasserstein space convex?

For the Gaussian family, the gradient flow in equ. (4) does not require the calculation of the Hessian of the log posterior density. However, the gradient flow of Gaussian mixture in equ. (10) does require the Hessian. As the calculation of the Hessian for high dimensional inference problem can be problematic and the proposed method itself is a first-order method, is it possible to only utilize the gradient information of the log-posterior density?

Some literature related to the Bures-Wasserstein space are missing, see [1, 2].

[1] Malagò, Luigi, Luigi Montrucchio, and Giovanni Pistone. "Wasserstein Riemannian geometry of Gaussian densities." Information Geometry 1.2 (2018): 137-179.

[2] Modin, Klas. "Geometry of matrix decompositions seen through optimal transport and information geometry." arXiv preprint arXiv:1601.01875 (2016).


**Limitations:**

Yes.

**Strengths And Weaknesses:**

Strength:
- The paper is well-motivated
- It provides an interesting interpretation of the variational Kalman filtering from the perspective of Wasserstein gradient flow.
- The convergence analysis of the proposed Bures-Wasserstein SGD is novel.
- The numerical results of the proposed method for Gaussian mixture seem promising.

Weakness:
- The theoretical contribution is not enough. For the Gaussian family, the analysis of the continuous-time dynamics seems to be a direct application of the convergence results of Wasserstein gradient flow. For the Gaussian mixture, although the author acknowledge that the problem is non-convex, some theoretical analysis of the proposed method can strengthen the paper. See details in the Questions.
- The structure of the paper can be improved. The algorithm for Gaussian mixture is of great interest in practice but the detailed description of that algorithm is not presented in the paper.
- The comparison of the proposed method for Gaussian mixture with other variational inference methods using Gaussian mixture is missing. Some experiments on Bayesian inference can strengthen the paper.

---

> ### Author Response · Authors · 2022-08-01
> **Response**
>
> Thank you for your constructive suggestions. Below, we address some of your points in turn.
>
> > For the Gaussian family, the analysis of the continuous-time dynamics seems to be a direct application of the convergence results of Wasserstein gradient flow. For the Gaussian mixture, although the authors acknowledge that the problem is non-convex, some theoretical analysis of the proposed method can strengthen the paper.
>
> Our mathematical analysis for the Gaussian case is indeed a direct application of Wasserstein theory that leverages the fact that the BW manifold is a totally geodesic subset of the Wasserstein space.   We believe that this constitutes novel and strong evidence that the well-developed theory of Wasserstein gradient flows provides the correct framework to design and analyze VI algorithms (note that it is the first justification of Särkkä’s heuristic which dates back to 2007). While previous work necessitates a clever ad-hoc square-root reparametrization to elicit convexity, the Wasserstein framework is completely natural and directly leads to a new algorithm for mixtures of Gaussians. A theoretical analysis for the landscape of mixtures of Gaussians is a notoriously hard problem that is inherent to parametrization rather than geometry (see below for more comments in this direction).
>
> > The structure of the paper can be improved. The algorithm for Gaussian mixture is of great interest in practice but the detailed description of that algorithm is not presented in the paper.
>
> The algorithm is a numerical implementation of the two coupled ordinary differential equations (11) and (12). The revision now emphasizes that they constitute the core of the algorithm. In turn, numerical implementation of the ODE and the expectations it contains may be performed following various standard methods: cubature rules and Runge–Kutta. These numerical strategies are somewhat classical and their detailed implementation is provided in Appendix J.2.
>
> > The comparison of the proposed method for Gaussian mixture with other variational inference methods using Gaussian mixture is missing. Some experiments on Bayesian inference can strengthen the paper.
>
> A Bayesian inference example has been considered for our Gaussian mixture model in Figure 1 and in Appendix J.3.1 with the logistic target. We have compared our algorithm with Laplace approximation in the single Gaussian case. In the mixture of Gaussians case, our experiments are a proof of concept that illustrates the practical relevance of Wasserstein gradient flows as a theoretical framework for VI. Detailed comparison with other methods for the mixture model is beyond the scope of this paper and left for future work.
>
> > The interpretation of the Gaussian mixture model as the measure on the Bures-Wasserstein space is interesting. I’m wondering if the target density itself is a Gaussian mixture, can the proposed algorithm exactly fit the posterior distribution?
>
> The reviewer has raised several questions regarding theoretical guarantees for the Gaussian mixture case. In fact, such difficulties are pervasive to Gaussian mixture modeling in which algorithmic guarantees for maximum likelihood estimation are very limited. In a way, the good interpretability of Gaussian mixtures comes at the cost of a difficult optimization landscape. Overcoming these algorithmic barriers is an active and ongoing area of research that is beyond the scope of this paper. Indeed, while we argue for Wasserstein gradient flows as a generic framework for VI, we do not claim that it can solve all problems. However, akin to expectation/maximization for likelihood maximization, it provides a novel heuristic that performs well in numerical experiments. In particular, it appears that with appropriate initialization it can indeed recover the target in the well-specified case.
>
> (response continued in next comment)

---

> > ### Author Response · Authors · 2022-08-01
> > **Response (cont.)**
> >
> > (continued from previous comment)
> >
> >
> > > For the Gaussian mixture, are there objective functionals other than the KL divergence, for instance, maximum mean discrepancy, to make the resulting problem on the measure of the Bures-Wasserstein space convex?
> >
> > This is an interesting research direction and other objectives have indeed been proposed in the VI literature, albeit not necessarily in the context of Wasserstein gradient flows (see, e.g., Daudel et al.). Note however, that convexity depends not only on the objective but also on the chosen parametrization (together with the geometry with which the parameter space is endowed). In particular, modifying the objective to MMD for example, may not be sufficient to obtain convexity. On the other hand, simple modifications of the parametrization can lead to convexity; for example choosing the mixing measure itself as a parameter makes the problem convex for the classical L^2 geometry but it is not amenable to good implementation strategies. In this convexity/implementability tradeoff, Wasserstein gradient flows strike a good balance because not only do they lead to tractable interacting particle systems but also to theoretical guarantees in specific cases (single particle with log-concave target).
> >
> > > For the Gaussian family, the gradient flow in equ. (4) does not require the calculation of the Hessian of the log posterior density. However, the gradient flow of Gaussian mixture in equ. (10) does require the Hessian. As the calculation of the Hessian for high dimensional inference problem can be problematic and the proposed method itself is a first-order method, is it possible to only utilize the gradient information of the log-posterior density?
> >
> > Hessian-free updates for Gaussian mixtures can also be obtained via integration by parts; it is presented in Appendix A.2 due to a lack of space. We have added a pointer in the main body.
> >
> > > Some literature related to the Bures-Wasserstein space are missing, see [1, 2].
> >
> > Thank you for these pointers. We will add them in our next revision.

---

> > > ### Comment · Reviewer_8QPd · 2022-08-04
> > > **Response to the authors**
> > >
> > > The authors address most of my concerns and I raise the score.

---

### Official Review · Reviewer_qdtG · 2022-07-11

**Rating:** 7
**Confidence:** 3
**Soundness:** 3 good
**Presentation:** 3 good
**Contribution:** 3 good

**Summary:**

The paper provides an analysis of variational inference algorithms based on Gaussian approximation of variational distributions. The paper exploits the theory around Bures-Wasserstein spaces of Gaussian measures to come construct a sequence of Gaussian distributions (or mixtures) that converge to the target measure, i.e., the variational approximation of the target distribution.
The paper uses Wasserstein gradient flows for their theoretical analysis and establishes that Saerkkae’s heuristic coincides with the sequence of distributions arising in the gradient flow of the KL divergence.
This analysis is extended to the use Gaussian mixture distributions.
Theoretical guarantees are derived and discussed in detail. The theoretical results rely on the assumption of log concave distribution, an assumption that is also made in the study of Langevin type algorithms.

**Questions:**

see above

**Limitations:**

Yes

**Strengths And Weaknesses:**

The theoretical contributions of the paper look sound and the work is well situated among other contributions in the field.

To add:
In the related work section it might be worthwhile to add reference to the work on sequential Monte Carlo algorithms.

Minor:
Line 272: generality?

Contributions:
- Add more numerical results in the main body of the paper? This would make the work stronger on spark practical applications

Line 335 : “ …. yields a new Gaussian particle method which appears to be significantly more powerful than classical particle methods.”
This is an overly strong statement that requires more justification. You are not explicitly discussing how your method is more powerful. This should happen both from an empirical and theoretical point of view (which you both don’t do).

---

> ### Author Response · Authors · 2022-08-01
> **Response**
>
> Thank you for your kind words and helpful suggestions. We will incorporate them into the revised manuscript.

---

### Author Response · Authors · 2022-08-01
**Comment on missing references**

Thank you to all reviewers for the helpful feedback. After submitting this manuscript, we became aware of a line of work (e.g., [1, 2]) which also obtains convex guarantees for Gaussian VI via the square root parameterization of the covariance matrix in the context of Bayesian logistic regression. We will revise our submission to include these references and provide a discussion of the placement of our paper in the context of this work.

Although this literature shows that quantitative guarantees for VI can also be obtained via alternative methods, we believe that our approach via Wasserstein gradient flows remains a natural and powerful framework for extensions to other settings, as we demonstrate via the family of mixtures of Gaussians.

[1] E. Challis and D. Barber. Gaussian Kullback–Leibler approximate inference. 2013.

[2] P. Alquier and J. Ridgway. Concentration of tempered posteriors and of their variational approximations. 2019.

---

### Public Comment · ~Qiang_Fu11 · 2023-03-29
**Question of formula (3)**

Dear authors,

I really enjoy reading this paper, but I’m a little confused about a notation in formula (3). Dose $\otimes$ in (3) mean kronecker product or tensor product? Seems like $\nabla V(X_t) \otimes (X_t-m_t)$ should be a matrix with the same dimension as $I$. Is this notation equivalent to $\nabla V(X_t) (X_t-m_t)^T$? Could you clarify that?

Thanks

---

> ### Public Comment · Authors · 2023-04-01
> **Response**
>
> Yes, you are correct, it means the same as the outer product of the two vectors.

---

> > ### Public Comment · ~Qiang_Fu11 · 2023-04-04
> > **Derivation of formula (3)**
> >
> > Thank you for the response!
> >
> > When you derive equation (3) by fokker-planck equation, you use the proof $\dot{m_t}=\partial_t \mathbb{E} x_t=\mathbb{E} \dot{x_t}$. But $\mathbb{E} x_t=\int x_t \pi_t(x_t)$. Shouldn't it be $\partial_t \mathbb{E} x_t=\int \dot{x_t} \pi_t(x_t)+\int x_t \partial_t \pi_t(x_t)=\mathbb{E} \dot{x_t}+\int x_t \partial_t \pi_t(x_t)$ instead of just $\mathbb{E} \dot{x_t}$?

---

> > > ### Public Comment · Authors · 2023-04-05
> > > **Response**
> > >
> > > No this is a misunderstanding: when you write $\mathbb E x_t = \int x_t \\, \pi_t(dx_t)$, here $x_t$ is a dummy variable (variable of integration) and does not actually depend on time. It would be clearer to write it as $\int x \\, \pi_t(dx)$.

---

### Public Comment · ~Mingxuan_Yi1 · 2023-10-10
**The purpose of the clip operator**

Hello. I really like this paper and I have one question regarding the clip operator in Algorithm 1, is it to ensure the smallest eigen value being greater than zero such that the covariance is always positive definite?

---

> ### Public Comment · Authors · 2023-10-11
> **Response**
>
> Yes; the clip operator truncates the largest eigenvalue of the covariance from above, which by Lemma 2 in the supplement leads to a control of the smallest eigenvalue of the covariance matrix along the iterations. An alternative algorithmic approach to control the eigenvalues of the covariance matrix is to use proximal gradient (see https://arxiv.org/abs/2304.05398).

---

> > ### Public Comment · ~Mingxuan_Yi1 · 2023-11-01
> > **An update**
> >
> > Hi, thanks for your comment. I recently uploaded a paper on Arxiv https://arxiv.org/pdf/2310.20090.pdf. That is about the standard BBVI with the reparameterization gradient (parameterizing Gaussians with scale matrices) can be viewed as the forward scheme to the Bures-Wasserstein gradient flow. This is because the Euclidean gradient $\nabla_S KL(q||p)$ w.r.t. the scale matrix $S$ is a horizontal vector under the Riemannian submersion $\pi(S)=SS^T=\Sigma$ (see section 4 in Bhatia et. al. https://arxiv.org/pdf/1712.01504.pdf)

---

### Meta-Review · Area_Chair_enBP · 2022-08-26

**Recommendation:** Accept
**Confidence:** Certain

**Metareview:**

This paper proposes a novel method for variational inference based on Wasserstein flows. The key contribution is perhaps the rigorous guarantees that are derived from an assumption of log-concavity. While the initial submission was unaware of some existing work on VI that derives guarantees from similar log concavity or smoothness assumptions, the proof strategy that is given uses novel technical methods, and thus is of interest in any case. Readers would benefit from a detailed discussion that can contextualize this work to previous work, which the authors have committed to doing.

**Award:**

No

---

### Decision · Program_Chairs · 2022-09-14

Accept